# Genetic and pharmacological evidence for kinetic competition between alternative poly(A) sites in yeast

Rachael Emily Turner[1], Paul F Harrison[1,2], Angavai Swaminathan[1], Calvin A Kraupner-Taylor[1], Belinda J Goldie[1], Michael See[1,2], Amanda L Peterson[3], Ralf B Schittenhelm[4], David R Powell[2], Darren J Creek[3], Bernhard Dichtl[5]*, Traude H Beilharz[1]*

[1]Development and Stem Cells Program, Monash Biomedicine Discovery Institute and Department of Biochemistry and Molecular Biology, Monash University, Melbourne, Australia; [2]Monash Bioinformatics Platform, Monash University, Melbourne, Australia; [3]Drug Delivery, Disposition and Dynamics, Monash Institute of Pharmaceutical Sciences, Monash University, Parkville, Australia; [4]Monash Proteomics & Metabolomics Facility, Department of Biochemistry and Molecular Biology, Monash Biomedicine Discovery Institute, Monash University, Melbourne, Australia; [5]School of Life and Environmental Sciences, Deakin University, Geelong, Australia

*For correspondence:
bernhard.dichtl@deakin.edu.au (BD);
traude.beilharz@monash.edu (THB)

**Competing interests:** The authors declare that no competing interests exist.

**Abstract** Most eukaryotic mRNAs accommodate alternative sites of poly(A) addition in the 3' untranslated region in order to regulate mRNA function. Here, we present a systematic analysis of 3' end formation factors, which revealed 3'UTR lengthening in response to a loss of the core machinery, whereas a loss of the Sen1 helicase resulted in shorter 3'UTRs. We show that the anti-cancer drug cordycepin, 3' deoxyadenosine, caused nucleotide accumulation and the usage of distal poly(A) sites. Mycophenolic acid, a drug which reduces GTP levels and impairs RNA polymerase II (RNAP II) transcription elongation, promoted the usage of proximal sites and reversed the effects of cordycepin on alternative polyadenylation. Moreover, cordycepin-mediated usage of distal sites was associated with a permissive chromatin template and was suppressed in the presence of an *rpb1* mutation, which slows RNAP II elongation rate. We propose that alternative polyadenylation is governed by temporal coordination of RNAP II transcription and 3' end processing and controlled by the availability of 3' end factors, nucleotide levels and chromatin landscape.

## Introduction

3' end processing is a critical step in eukaryotic messenger RNA (mRNA) maturation. This two-step process involves co-transcriptional endonucleolytic cleavage of the pre-mRNA transcript and the subsequent addition of a non-templated polyadenosine (poly(A)) tail by poly(A) polymerase (*Colgan and Manley, 1997*). Cleavage site positioning is defined by a series of sequence elements within the pre-mRNA transcript that are recognised by the multi-subunit complexes that form the 3' end processing machinery. These cleavage and polyadenylation factors are highly conserved between eukaryotes; however, some differences in proteins and organisation exist between organisms. In the budding yeast *Saccharomyces cerevisiae*, three subcomplexes make up the core 3' end processing machinery. These include cleavage and polyadenylation factor (CPF), cleavage factor IA (CFIA) and cleavage factor IB (CFIB) (*Zhao et al., 1999*; *Mandel et al., 2008*).

**Figure 1.** Mutation of the yeast cleavage and polyadenylation machinery subunits confer changes in cleavage site choice. (**A**) Schematic of the yeast core cleavage and polyadenylation machinery and their interactions with the nascent mRNA strand. Cleavage and polyadenylation factor (CPF), Cleavage factor IA (CFIA) and Cleavage factor IB (CFIB) are involved in canonical 3' end formation whereas the Nrd1-Nab3-Sen1 (NNS) complex represents an alternative cleavage pathway utilised by snRNAs, snoRNAs, CUTs and some mRNAs. (**B**) Yeast strains, as indicated, were serially diluted 10-fold onto YPAD rich media plates and grown at 25°C or 37°C for 2 days. (**C**) Integrative Genomics Viewer (IGV) (*Robinson et al., 2011*) representation of PAT-seq reads aligned to the 3' end of the *SUB2* gene. Indicated yeast strains were grown at 25°C until OD$_{600}$ 0.6 then switched to 37°C for 1 hr. Peaks indicate nucleotide depth of coverage by sequencing reads and data ranges for each sample are shown on the left. The ratio of these peaks is used to calculate 3' end shifts. Schematic of 3' UTR length for each peak is shown below. (**D**) Global 3' end shift effects of genes observed in mutants of the indicated 3' end processing factor subunits. Values are relative to the wild-type strain W303 at 37°C using PAT-seq data. Each dot corresponds to one gene. A positive effect size indicates a general lengthening of the 3' UTR in the mutant relative to the wild-type, whereas a negative effect size indicates shifts towards shorter 3' UTRs. Grey dots are the estimated effect size. Where this is significant, a corresponding dot shows the 'confect', a confident inner bound of this effect size (FDR < 0.05) found using the Limma and TopConfects R packages. Red indicates significant lengthening in the mutant whereas blue indicates significant shortening. The total number of detected genes (grey dots) and the number of significant shifts (red and blue dots) are indicated for each strain. Log$_2$(RPM) is the logarithmic average number of reads per million. Data is representative of two biological replicates and is available in the *Figure 1—source data 1*.

*Figure 1 continued on next page*

*Figure 1 continued*

The online version of this article includes the following source data for figure 1:

**Source data 1.** 3′end processing mutant allele PAT-seq end shift values.

Together, these interact with the nascent transcript to promote both cleavage and polyadenylation (*Figure 1A*, *Graber et al., 1999*).

The 3′ end processing signals involved in cleavage site recognition in yeast are generally more divergent than their mammalian counterparts and lack strong consensus sequences (*Graber et al., 1999*). However, in *S. cerevisiae*, four main sequence elements are associated with cleavage site positioning. Firstly, there is an A-rich positioning element (PE) located 10–30 nucleotides upstream of the cleavage site (*Tian and Graber, 2012*). This is the equivalent of the mammalian polyadenylation signal (PAS) and has the same optimal sequence of AAUAAA though this motif is less conserved in yeast (*Zhao et al., 1999*). Secondly, a UA-rich efficiency element (EE) occurs upstream of the PE sequence. The optimal sequence for the EE is UAUAUA and is generally found 25–40 nucleotides away from the cleavage site (*Tian and Graber, 2012*). This sequence is not required for cleavage but improves the efficiency of the reaction and is important in site selection (*Guo and Sherman, 1996*). Lastly, U-rich elements exist either downstream (DUE) or upstream (UUE) of the cleavage site. Recombinant Yth1, Ydh1/Cft2, and Yhh1/Cft1 (all CPF subunits) as well as purified CPF interact with these U-rich elements (*Barabino et al., 2000*; *Dichtl and Keller, 2001*; *Dichtl et al., 2002b*) and mutation of these elements can result in reduced cleavage activity (*Dichtl and Keller, 2001*). Hrp1/Nab4, the sole CFIB subunit, interacts with the EE (*Kessler et al., 1997*; *Chen and Hyman, 1998*; *Valentini et al., 1999*). Rna15 (a CFIA subunit) has been found to interact with the PE motif in vitro (*Gross and Moore, 2001*), whereas in vivo mapping of factors on the nascent transcript suggested that binding of CFIA occurs downstream of the cleavage site (*Baejen et al., 2014*) similar to the CstF complex in the mammalian system (*Chan et al., 2011*; *Mandel et al., 2008*; *Martin et al., 2012*). Thus, the overall assembly of the 3′ end formation machinery on the pre-mRNA substrate appears to be conserved between yeast and human (*Baejen et al., 2014*).

Pre-mRNA 3′ end formation is physically and functionally coupled to RNAP II transcription. Assembly of 3′ end factors on an emerging poly(A) site is accompanied by interactions with the carboxy terminal domain (CTD) of Rpb1, the largest subunit of RNAP II (*McCracken et al., 1997*). The CFIA subunit Pcf11 plays a central role in transmitting poly(A) site recognition to elongating RNAP II, triggering termination of transcription (*Zhang et al., 2005*; *Sadowski et al., 2003*; *Barillà et al., 2001*).

Many genes possess more than one potential site at which cleavage and polyadenylation may occur. This occurrence, termed alternative polyadenylation (APA), is pervasive in all studied eukaryotes and appears in up to 70% of human and yeast genes (*Derti et al., 2012*; *Ozsolak et al., 2010*). Use of such alternate cleavage sites has the potential to impact the stability, localisation and translational efficiency of an mRNA transcript. This has been linked to biological processes such as development and proliferation as well as diseases including cancer (*Mayr and Bartel, 2009*; *Ji et al., 2009*; *Sandberg et al., 2008*). However, the mechanisms that govern cleavage site choice are still under investigation.

Here, we used mutant alleles of yeast core cleavage and polyadenylation factors to probe their roles in alternative polyadenylation. Mutant phenotypes were linked to global preferred usage of distal cleavage sites and 3′ UTR lengthening irrespective of subcomplex membership. Similarly, the adenosine analogue cordycepin (3′ deoxyadenosine) was shown to interfere with correct 3′ end processing and promote APA comparable to inactivation of the cleavage and polyadenylation machinery. This was linked to increased cellular nucleotides, and an enhanced transcription elongation rate in response to the drug treatment. In addition, the occurrence of APA following cordycepin treatment correlated with the size of intergenic regions and the size of nucleosome depleted regions after the stop codon in convergent genes. We propose that APA in yeast employs a balance between cleavage site strength, 3′ end processing factor availability, and transcriptional rate to establish a kinetic control of poly(A) site choice.

## Results

### Mutations within the core cleavage and polyadenylation machinery promote lengthening of 3' UTRs

A compendium of *S. cerevisiae* mutants with defects in the cleavage and polyadenylation machinery has been collated from members of the yeast 3' end formation community (*Table 1*). This included strains with mutated subunits across the three core 3' end processing subcomplexes CFIA, CFIB and CPF, as well as the non-canonical Nrd1-Nab3-Sen1 (NNS) complex (*Figure 1A*). This repository combined with modern transcriptome-wide analyses has provided a unique opportunity to further understand how 3' end choices are globally regulated in yeast. We, therefore, sought to explore the role of individual cleavage factor subunits in cleavage site selection.

The majority of the cleavage and polyadenylation factors are essential for life necessitating the use of primarily temperature-sensitive alleles (*Figure 1B*). The phenotypes associated with mutant strains analysed here have been well described in the literature (*Table 1*). This includes primarily loss-of-function mutations, displaying distinct phenotypes in various steps of mRNA synthesis including pre-mRNA cleavage, pre-mRNA polyadenylation, cleavage site recognition, RNA binding, or transcription termination. For consistency, irrespective of the severity of the mutation, duplicates of each strain were grown in rich media at 25°C until an $OD_{600}$ of 0.6 was reached, then switched to a restrictive temperature (37°C) for 1 hr before harvesting.

A representative set of seven mutants was chosen for full transcriptome-wide APA analysis. This included three mutants from the CFIA complex, *rna14-1*, *pcf11-2*, and *clp1-pm*, a mutant of the single protein factor CFIB, *nab4-1*, and three mutants from the CPF complex, *ysh1-13*, *fip1-1*, and *pap1-1*. With the exception of *clp1-pm*, these were temperature-sensitive mutants (*Figure 1B*). Gene expression and cleavage site choice were analysed using the targeted poly(A) tail sequencing approach Poly(A)-Test Sequencing (PAT-seq) (*Harrison et al., 2015*). This is a 3' end-focused deep sequencing technique that involves biotin-labelled anchor oligonucleotide primers, limited digestion and gel size selection to enrich for the final 200 bp of polyadenylated transcripts as input into Illumina compatible sequencing libraries.

Sequencing data was processed using the tail-tools pipeline (http://rnasystems.erc.monash.edu/software/; *Harrison et al., 2015*). Sequencing reads were mapped to genes within the *S. cerevisiae* genome and the 3' end of transcripts were analysed for differences in positioning relative to the wild-type at 37°C. *Figure 1C* shows the 3' UTR shifts associated with inactivation of the 3' end processing machinery for the representative gene *SUB2*. Peak height reflects the number of transcripts ending at a certain position in the aligned sequence. Mutation of the *RNA14*, *PCF11*, *HRP1/NAB4*, *YSH1*, *FIP1*, and *PAP1* genes caused a switch to more distal cleavage site usage and longer 3' UTRs. This change occurred to the greatest extent in the *rna14-1* and *pcf11-2* mutants where reads mapping to the proximal site were rarely found. Little change was noticeable for the *clp1-pm* mutant, which contains point mutations in the P-loop motif (K136A and T137A) suggesting that ATP binding by Clp1 was dispensable for poly(A) site selection. Consistently, the mutations have previously been shown to have limited impact on pre-mRNA 3' end formation and termination in vitro (*Holbein et al., 2011*).

Of the 7091 genomic annotations identified by PAT-seq for our samples, 3511 were found to have two or more poly(A) sites within their 3' UTRs. Shift scores estimating the extent of 3' end shifting at 37°C in the seven mutants relative to the wild-type strain W303 were calculated for each of these genes. Effect size shift scores range from −1 to 1 with negative values indicating a shift to a more proximal cleavage site in the mutant and a positive shift score indicating a switch to a more distal cleavage site. An inner confidence bound, or 'confect', on these effect size scores was given for genes with significantly non-zero shifts (FDR < 0.05) using the topconfects R package (*Harrison et al., 2019*). Significant 3' UTR shifts were found in all seven mutants compared to the wild-type (*Figure 1D*). Excluding *clp1-pm*, 422 genes significantly changed cleavage sites in all mutants. The majority of observed shifts generated longer 3' UTRs upon mutation of the cleavage machinery indicating that functional attenuation of the individual subunits generally promoted an increase in 3' UTR length. *ysh1-13* had the highest number of significant shifts with 2486 genes significantly changing their cleavage site. Of these, 2457 switched to longer 3' UTRs compared to wild-type whilst only 29 became shorter. Ysh1, an endonuclease, is assumed to be the subunit responsible for cleaving the pre-mRNA transcript (*Ryan et al., 2004*; *Mandel et al., 2006*; *Garas et al.,*

**Table 1.** Yeast Strains.

| Strain | Genotype | Relevant mutation if known | Reference |
|---|---|---|---|
| W303 | MATa leu2-3,112 his3-11,15 trp1-1 can1-100 ade2-1 ura3-1 | Wild type | Thomas and Rothstein, 1989 |
| BY4741 | MATa leu2Δ0 his3Δ1 ura3Δ0 met15Δ0 | Wild type | Brachmann et al., 1998 |
| rna14-1 | MATa leu2-3,112 his3-11,15 trp1-1 ade2-1 ura3-1 rna14-1 | T nucleotide insertion at 1986 causing a premature stop codon (replacement of last 16 amino acids with FK) | Bloch et al., 1978 (named cor1-1); Minvielle-Sebastia et al., 1994, Rouillard et al., 2000 |
| rna15-1 | MATa leu2-3,112 his3-11,15 trp1-1 ade2-1 ura3-1 rna15-1 | L214P | Bloch et al., 1978 (named cor2-1); Minvielle-Sebastia et al., 1994, Qu et al., 2007 |
| pcf11-2 | MATa leu2-3,112 his3-11,15 trp1Δ can1-100 ade2-1 ura3-1 pcf11-2 | E232G, D280G, C424R, S538G, F562S, S579P | Amrani et al., 1997, Sadowski et al., 2003 |
| pcf11-13 | MATa leu2-3,112 his3-11,15 trp1Δ can1-100 ade2-1 ura3-1 pcf11-13 | D68A, S69A, I70A (CID region) | Sadowski et al., 2003 |
| clp1-pm | MATa leu2-3,112 his3-11,15 trp1-1 can1-100 ade2-1 ura3-1 clp1-pm | K136A, T137A (P-loop motif) | Ramirez et al., 2008 |
| nab4-1 | MATa ura3-52 nab4-1 | N167D, F179Y, P194H, Q265L | Minvielle-Sebastia et al., 1998 |
| nab4-7 | MATa ura3-52 nab4-7 | S64R, K92M, T125A, I163T | Minvielle-Sebastia et al., 1998 |
| yhh1-3 | MATa leu2-3,112 his3-11,15 trp1-1 can1-100 ade2-1 ura3-1 yhh1-3 | L302P, N716S, N762S, Y766C, K933R, D1070E, N1136D | Dichtl et al., 2002b |
| ysh1-13 | MATa leu2-3,112 his3-11,15 trp1Δ ade2-1 ura3-1 TRP1::ysh1 [ysh1-3-HIS3-CEN] | V235A, N685H, D695V, E723V, R763G | Garas et al., 2008 |
| yth1-1 | MATa leu2-3,112 his3-11,15 trp1Δ ade2-1 ura3-1 yth1::TRP1 [CEN4-ADE2-yth1-1] | frameshift at 154 in causing a premature stop codon (deletion of last 55 amino acids) | Barabino et al., 1997 |
| yth1-4 | MATa leu2-3,112 his3-11,15 trp1Δ ade2-1 ura3-1 yth1::TRP1 [CEN4-ADE2 -yth1-4] | W70A | Barabino et al., 2000 |
| fip1-1 | MATa leu2-3,112 his3-11,15 trp1-1 ura3-1 fip1-1 | L99F, Q216X (deletion of last 111 amino acids) | Preker et al., 1995, Preker et al., 1997 |
| pap1-1 | MATa ade1 ade2 lys2 gal1? ura3-52 pap1-1 | | Patel and Butler, 1992 |
| pta1-1 | MATa leu2-3,112 his3-11,15 trp1-1 ade2-1 ura3-1 pta1-1 | Premature stop codon | O'Connor and Peebles, 1992, Preker et al., 1997 |
| pfs2-1 | MATa leu2-3,112 his3-11,15 trp1Δ ade2-1 ura3-1 pfs2::TRP1 [pfs2-1-LEU2-CEN] | | Ohnacker et al., 2000 |
| mpe1-1 | MATα leu2-3,112 his3-11,15 trp1-1 ade2-1 ura3-1 mpe1-1 | F9S, Q268K, K337F, K354X (deletion of last 87 amino acids) | Vo et al., 2001 |
| ref2-2 | MATa leu2-3,112 his3-11,15 trp1-1 can1-100 ade2-1 ura3-1 ref2-2 | | Dheur et al., 2003 |
| pti1-2 | MATa leu2-3,112 his3-11,15 trp1-1 can1-100 ade2-1 ura3-1 pti1-2 | | Dheur et al., 2003 |
| ssu72-2 | MATa leu2-3,112 his3-11,15 trp1-1 ade2-1 ura3-1 ssu72-2 | R129A | Pappas and Hampsey, 2000, Dichtl et al., 2002a |
| glc7-5 | MATa leu2-3,112 his3-11,15 can1-100 ade2-1 ura3-1 glc7::LEU2 trp1:: glc7-5 | F226L | Peggie et al., 2002 |
| swd2-2 | MATa leu2Δ0 his3Δ1 ura3Δ0 met15Δ0 swd2-2 | F14P, C27G, F41I, K185E, S253L, C257R | Dichtl et al., 2004 |
| Δsyc1 | MATa leu2Δ0 his3Δ1 ura3Δ0 met15Δ0 syc1Δ | Whole gene deletion | Zhelkovsky et al., 2006 |
| nrd1-5 | MATa ura3-52 trp1-1 ade2-1 leu2-3,112 his3-11,15 lys2-Δ2 can1-100 met2-Δ1 nrd1-5 | G368V (RNA recognition motif) | Steinmetz and Brow, 1996 |
| nrd1-ΔCID | MATa ura3-52 trp1-1 ade2-1 leu2-3,112 his3-11,15 lys2-Δ2 can1-100 met2-Δ1 nrd1-ΔCID | deletion of amino acids 39–169 (CTD region) | Steinmetz and Brow, 1998 |
| nab3-11 | MATa leu2-3,112 his3-11,15 trp1-1 can1-100 ade2-1 ura3-1 nab3-11 | F371L, P374L (RNA recognition motif) | Conrad et al., 2000 |

*Table 1 continued on next page*

*Table 1 continued*

| Strain | Genotype | Relevant mutation if known | Reference |
|---|---|---|---|
| *sen1-1* | MAT**a** *leu2-3,112 his3-11,15 trp1-1 ade2-1 ura3-1 sen1-1* | G1747D (helicase domain) | *Winey and Culbertson, 1988*, *Mischo et al., 2011* |
| *rbp1-1* | MAT**α** *ura3-52 leu2-3,112 rpb1-1* | | *Nonet et al., 1987* |

*2008*) and *ysh1* mutants may cause the proximal site to be skipped due to inefficiency of the actual cleavage reaction.

Despite more genes being affected in the *ysh1-13* strain, *rna14*-1, *pcf11-2,* and *nab4-1* strains all showed greater extents of shifting in impacted genes with absolute average confect values of 0.150, 0.125, and 0.093, respectively, compared to 0.086 in *ysh1-13*. The extent of the shift reflects the ratio of proximal to distal site usage. This suggests a key role for the CFIA and CFIB complexes in cleavage site selection. The role of these subunits in alternative polyadenylation may be attributed to a decrease in interactions with the poly(A) site sequences. It is also important to note that though the extent of switching varies in different mutant strains, this may be due to the severity of the mutation on protein function rather than an actual difference in the mechanisms of cleavage site recognition. Overall, inactivation of the yeast 3′ end processing machinery appears to promote switching to more distal cleavage sites and higher expression of longer 3′ UTR with little specificity for individual subunits.

To ensure the cleavage factor mutants examined were indeed representative of the general effect 3′ end processing machinery impairment has on APA, a subset of shifts was analysed in the full panel of mutants. This was done using the multiplexed poly(A) test (mPAT) method (*Beilharz et al., 2017*), which is a targeted RNA sequencing adaptation of the extension poly(A) test (ePAT) assay (*Jänicke et al., 2012*) compatible with the Illumina MiSeq platform. The technique utilises a nested PCR to sequentially incorporate P5 and P7 extensions onto gene-specific PCR amplicons for sequencing on the Illumina flow-cell.

Forty-one genes were chosen for APA analysis, 39 of which were shown to significantly undergo alternative polyadenylation in the previous PAT-seq experiment. Cleavage site choice in each mutant strain was compared to that of the wild-type W303 at 37°C and shift scores calculated for each gene (*Figure 2A*). For the CFIA, CFIB, and CPF complexes, mutation of all but two subunits caused changes to cleavage site choice in at least a subset of the chosen genes. This was generally a switch to a more distal cleavage site and longer 3′ UTR usage. However, very little shifting was seen for the *clp1-pm* and *Δsyc1* mutants. Syc1 is the only non-essential subunit of the core cleavage and polyadenylation complexes (*Figure 1B*, *Zhelkovsky et al., 2006*) indicating that its role in cleavage site choice may be negligible.

In an attempt to further dissect functional requirements in APA, we analysed additional mutant alleles of selected factors. Reverse cumulative distribution plots of the 3′ ends of read alignments were used to visualise the percentage of reads terminating at each cleavage site of the gene *OM45* (*Figure 2B*). *nab4-1* and *nab4-7* mutants display defects in in vitro polyadenylation but function in in vitro cleavage (*Minvielle-Sebastia et al., 1998*). In our in vivo analyses both alleles were defective in *OM45* APA demonstrating a correlation with polyadenylation but not cleavage activity. The requirement for Hrp1/Nab4 in APA likely involves its role in poly(A) site selection (*Minvielle-Sebastia et al., 1998*). In vitro, the *yth1-1* and *yth1-4* alleles both have defects in the polyadenylation step of 3′ end formation but only *yth1-4*, and not *yth1-1*, is defective in 3′ end cleavage (*Barabino et al., 2000*). Additionally, in vivo APA defects did not correlate with cleavage activity, since both *YTH1* alleles showed comparable defects in APA (*Figure 2B*). The *pcf11-2* allele is defective in both cleavage and polyadenylation in vitro, while the *pcf11-13* allele is not (*Sadowski et al., 2003*). However, the latter contains mutations within the domain that interacts with the carboxyl-terminal domain (CTD) of Rpb1, the largest subunit of RNAP II (*Barillà et al., 2001*; *Sadowski et al., 2003*) and has formerly displayed defects in transcription termination (*Sadowski et al., 2003*). As such our APA results indicated that the interaction between Pcf11 and the transcriptional machinery was integral to APA in vivo. Overall, the results shown in *Figure 2B* suggested that defective in vitro cleavage activity was not indicative of defective APA in vivo. In contrast, defects in APA correlated with defective in vitro polyadenylation and defective interaction with the RNAP II CTD.

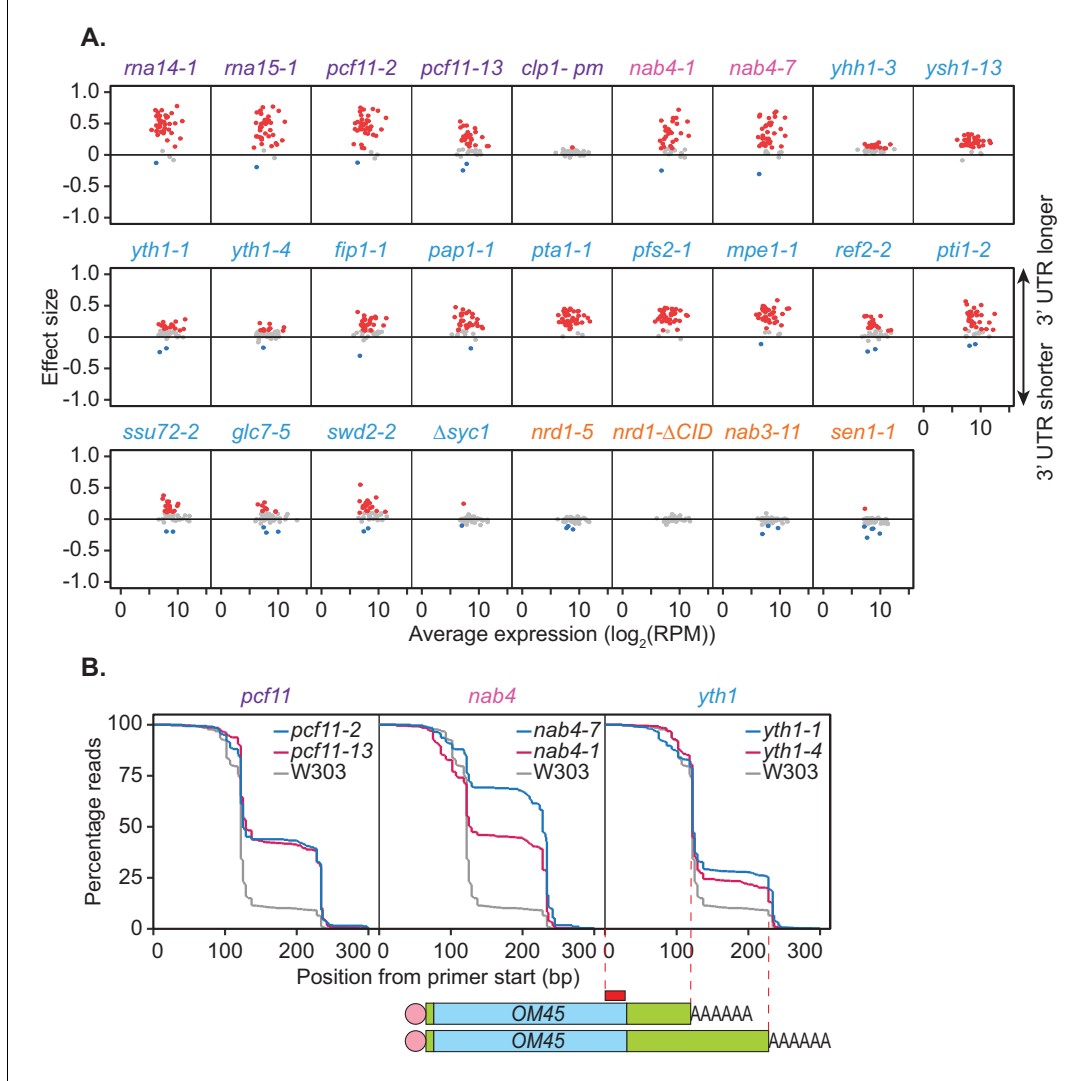

**Figure 2.** Mutations within the yeast cleavage and polyadenylation machinery subunits promote lengthening of mRNA 3' UTRs. (**A**) 3' end shift effects of genes observed upon mutation of the 3' end processing factor subunits. Values are relative to the wild-type strain W303 at 37℃ using mPAT data. Each dot corresponds to a one gene as above. Grey dots indicate an effect size between −0.1 and 0.1, red dots indicate an effect size greater than 0.1 and blue dots indicate an effect size less than −0.1. Data is representative of two biological replicates. (**B**) Comparison of 3' end shifting in different alleles of the same 3' end processing factor. Reverse cumulative distribution plots show raw templated nucleotide length of reads aligned to the gene *OM45* from mPAT data, starting from the forward primer position as illustrated (red box) in the schematic. Mutant alleles for a single cleavage factor are shown within the same plot in comparison to the wild-type strain W303.

Impairment of the NNS complex appeared to have little impact on cleavage site selection in this analysis. However, as only a subset of protein-coding genes require the NNS complex for transcriptional termination (*Steinmetz et al., 2006*; *Schulz et al., 2013*; *Steinmetz et al., 2001*), genes in which this complex plays a role may have been excluded in the mPAT analysis. To address this possibility, global APA was analysed in one mutant strain for each of the NNS subunits using the PAT-seq method as above (*Figure 3A*). *nrd1-5* and *nab3-11* had little to no defect in cleavage site choice with only 16 and 1 genes, respectively, shifting significantly compared to the wild-type. Interestingly, reduced function of the helicase Sen1 caused significant switching to shorter 3' UTRs for 676 genes. Of 411 genes that significantly switched cleavage site in both *sen1-1* and *pcf11-2*, 81.5% were in opposing directions (*Figure 3B,C*). Therefore, it appears that Sen1 activity was required to antagonise cleavage site choice by the core 3' end processing factors at a subset of genes. This was consistent with the previous report that a *sen1-E1597K* mutant causes premature termination and

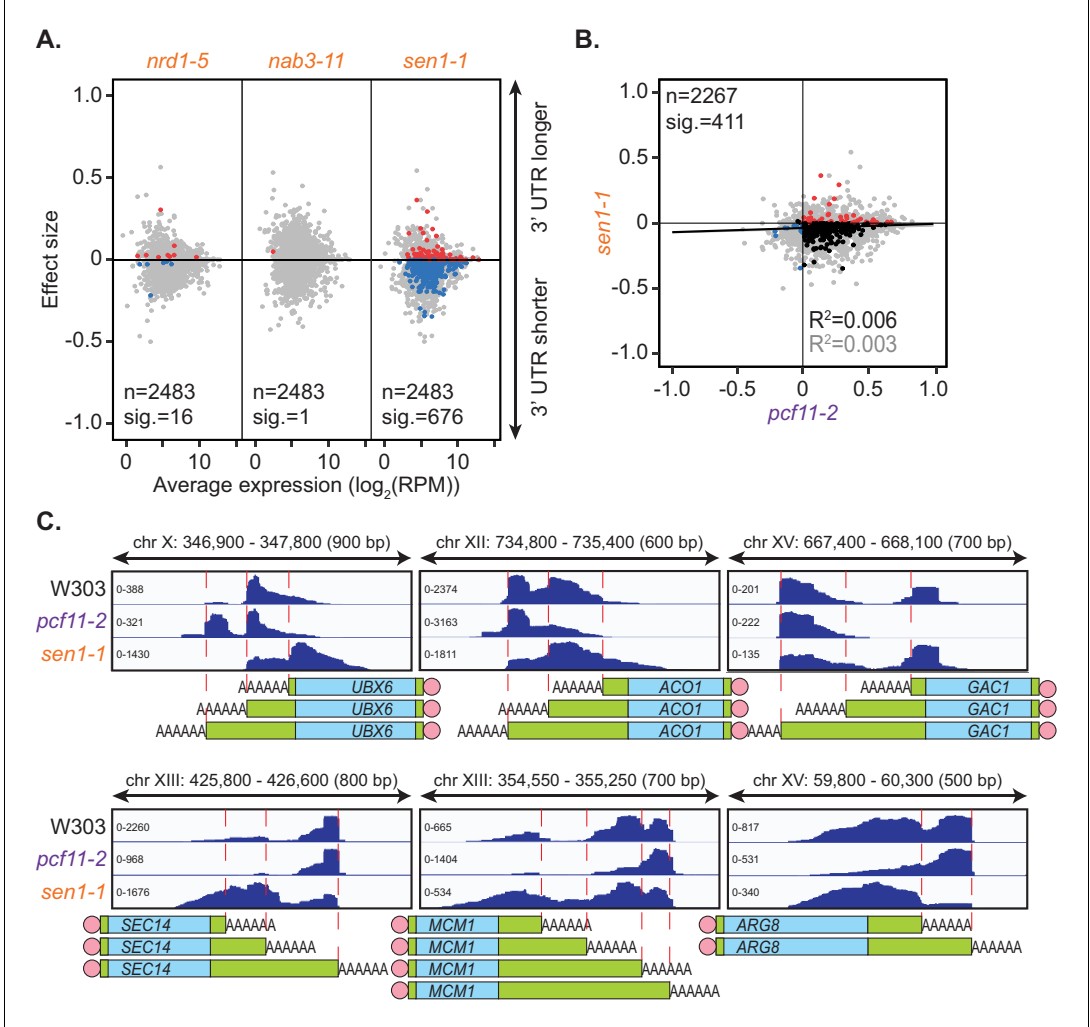

**Figure 3.** Sen1 mutation promotes 3' UTR shortening. (A) 3' end shift effects of genes observed upon mutation of the NNS subunits. Values are relative to the wild-type strain W303 at 37°C using PAT-seq data as in *Figure 1D*. Data is representative of two biological replicates and is available in the *Figure 3—source data 1*. (B) Comparison of 3' end shift effects of genes upon mutation of *PCF11* (x-axis) and *SEN1* (y-axis). *pcf11-2* and *sen1-1* shifts relative to their respective wild-type control at 37°C were compared. Grey dots indicate the estimated shift effects for all analysed features in common (n = 2267), black, red and blue dots indicate the significant shifts (n = 411, FDR < 0.05). Red indicates significant lengthening, blue indicates significant shortening and black indicates APA in opposite directions in both mutants. Lines show the linear regression for effect (grey) and confect (black) values respectively. (C) IGV representation of PAT-seq results aligned to the 3' end of the indicated genes in W303, pcf11-2, and sen1-1 yeast strains at 37°C. Peaks indicate nucleotide depth of coverage by sequencing reads and data ranges for each sample are shown on the left. Schematic of 3' UTR length for each peak is shown below.

The online version of this article includes the following source data for figure 3:

**Source data 1.** NNS mutant allele PAT-seq end shift values.

shortening of the 3' UTR of the *NPL3* transcript (*Steinmetz et al., 2006*). We found that no significant changes in transcript levels were associated with the APA events in *sen1-1* strains (data not shown); however, a significant increase in gene coding region length was observed for *sen1-1* responsive genes (1270.5 bp in *sen1-1* APA genes and 986 bp in no APA genes, p-value=$9.174 \times 10^{-9}$). Overall, our observations are consistent with a wider role for this RNA helicase in antitermination of protein coding genes, contrasting its previously described roles in non-coding RNA termination (see Discussion).

In sum, these genetic data indicate that APA hinges on the level of functional machineries for cleavage and polyadenylation. However, given that each temperature sensitive allele has a unique

sensitivity and severity and therefore a different degree of apparent impact on 3' end choice, we next sought to explore the impact of cordycepin as a possible pharmacological driver of APA.

## Cordycepin induces lengthening of 3' UTRs

Cordycepin, or 3' deoxyadenosine, is an adenosine analogue that causes chain termination when incorporated into RNA during synthesis. We have previously demonstrated that independently of this role, cordycepin can interfere with normal 3' end formation for several genes in yeast (*Holbein et al., 2009*). We therefore further examined the effect of cordycepin on cleavage site choice genome-wide.

Wild-type BY4741 yeast cells were treated with a non-lethal dose of cordycepin (20 µg/ml) for up to 40 min. Alternative polyadenylation was assessed using the PAT-seq method over a time course of drug treatment. Similar to depletion of the canonical cleavage and polyadenylation machinery, cordycepin treatment promoted use of a distal cleavage site for the gene *SUB2* (*Figure 4A*). Significant switching to alternative cleavage sites was observed for 1959 genes with the majority changing to longer 3' UTRs (*Figure 4B*). Therefore, cordycepin treatment generally caused a switch to more distal cleavage sites. Of note, any RNA with cordycepin incorporated will lack a 3' hydroxyl group and will therefore not be detectable by the PAT-seq assay. However, such transcripts are expected to be unstable and quickly targeted for degradation if premature termination occurs within the coding region of the gene (*Kamieniarz-Gdula and Proudfoot, 2019*).

How cordycepin interferes with cleavage site choice remains unknown. Genes affected by cordycepin treatment were therefore compared to those affected by mutations within the cleavage and polyadenylation machinery (*Figure 4—figure supplement 1*). The pattern of cleavage site switching in cordycepin treated cells was most similar to that of *pcf11-2* and *ysh1-13*, with 1391 and 1610 genes, respectively, being affected both by the drug and the mutation. However, there was also a group of 153 genes that had significant APA occurring only in the cordycepin-treated samples. This variation suggested that cordycepin did not alter cleavage site usage simply by interacting with or disabling the cleavage machinery. This is consistent with our observation above that mutants defective for in vitro cleavage retained proper APA in vivo (*Figure 2B*). Importantly, the genes that did significantly switch cleavage site in both cordycepin treated cells and the 3' end factor mutants, did so in the same direction with 1365 (98.1%) and 1592 (98.9%) of genes in quadrants 2 and 3 for cordycepin compared to *pcf11-2* and *ysh1-13,* respectively (*Figure 4C,D*). Overall, we conclude that cordycepin had a similar effect on cleavage site choice as mutations in the 3' end processing machinery but that it is unlikely to cause alternative polyadenylation through direct interaction with the 3' end formation factors.

Cordycepin treatment, however, had an impact on the expression of the cleavage and polyadenylation machinery (*Figure 4—figure supplement 2A*). The majority of the CPF complex subunits had decreased RNA expression after 40 min of drug treatment, whereas the CFIA and CFIB encoding genes *PCF11*, *RNA14,* and *HRP1/NAB4* showed increased expression. This may indicate the cell's attempt to return to wild-type cleavage positions, as these subunits were seen to play a key role in cleavage site selection. In addition, the NNS subunits *NRD1* and *SEN1* were upregulated. This might reflect a need to process the elevated level of cryptic unstable transcripts (CUTs) seen following cordycepin treatment (*Holbein et al., 2009*), or suggest a compensatory mechanism whereby cells under 3' end stress co-opt an alternative pathway to appropriately cleave mRNA transcripts (*Rondón et al., 2009*).

Cordycepin treatment also provided a convenient method to assess the implications of 3' UTR lengthening on translational efficiency within yeast. Polysome profiling of cells treated with cordycepin for 40 min demonstrated a decrease in polysome formation and, therefore, translation in response to cordycepin treatment (*Figure 4—figure supplement 2B*). This change could be due to stress from cordycepin's antifungal effect (*Sugar and McCaffrey, 1998*) or the global increase in 3' UTR length causing translation repression. Proteomic analysis of treated cells showed that this did not majorly transfer to global protein expression alterations with only five proteins significantly changing in expression (q-value 0.05, fold change > |1|) following cordycepin treatment when assessed using data-independent acquisition (DIA) mass spectrometry (*Figure 4—figure supplement 2C*). However, major changes in protein expression were not expected within this short time period of cordycepin treatment, and the lack of significant results may in part reflect the inherent inability of DIA mass spectrometry to assess newly translated proteins in isolation. Of note, the

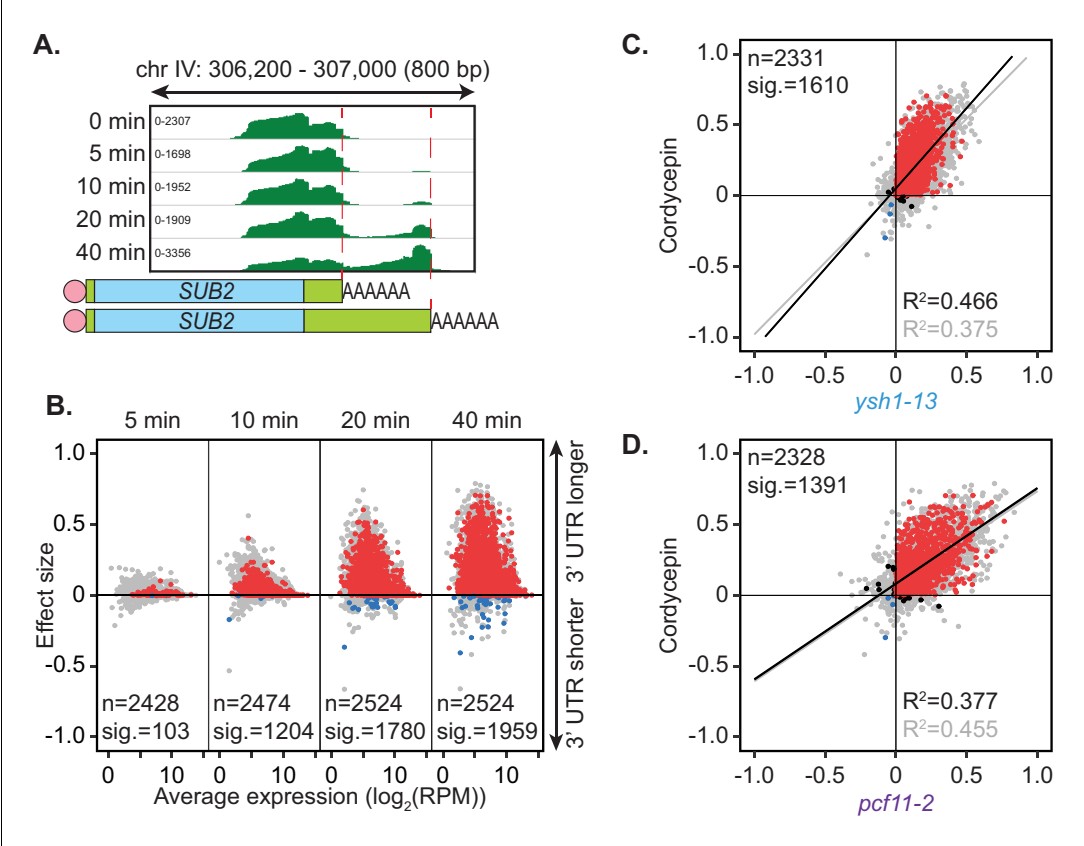

**Figure 4.** Cordycepin induces lengthening of 3' UTRs in yeast cells. BY4741 cells were treated with 20 μg/ml cordycepin for 0, 5, 10, 20, or 40 min and changes to 3' UTR length were observed. (**A**) IGV representation of PAT-seq reads aligned to the 3' end of the *SUB2* gene. Peaks indicate nucleotide depth of coverage by sequencing reads and data ranges for each sample are shown on the left. Schematic of 3' UTR length for each peak is shown below. (**B**) 3' end shift effects of genes observed upon cordycepin treatment. Values are relative to untreated cells using PAT-seq data as for *Figure 1D* with positive values indicating lengthening following drug treatment. Data is representative of two biological replicates and is available in the *Figure 4—source data 1*. (**C**) Comparison of 3' end shift effects of genes upon mutation of *YSH1* (x-axis) and after cordycepin treatment (y-axis). *ysh1-13* shifts were compared to W303 at 37°C and cordycepin treatment for 40 min was compared to untreated BY4741 cells at 30°C. Grey dots indicate the estimated shift effects for all analysed features in common (n = 2331), black, red, and blue dots indicate the significant shifts (n = 1610; FDR < 0.05). Red indicates significant lengthening, blue indicates significant shortening and black indicates APA in opposite directions for both the mutant and cordycepin treated cells. Lines show the linear regression for effect (grey) and confect (black) values, respectively. (**D**) Comparison of 3' end shift effects of genes upon mutation of *PCF11* (x-axis) and after cordycepin treatment (y-axis) as for C but with *pcf11-2* shifts compared to W303 at 37°C. The total number of analysed features in common was 2328 and the number of significant shifts was 1391 (FDR < 0.05).

The online version of this article includes the following source data and figure supplement(s) for figure 4:

**Source data 1.** Cordycepin treatment PAT-seq end shift values.

**Source data 2.** Cordycepin proteomics data.

**Figure supplement 1.** Cordycepin treatment and 3' end processing factor mutant alleles promote alternative polyadenylation in subsets of genes.

**Figure supplement 2.** Cordycepin treatment alters cleavage factor RNA expression and cellular translation.

cleavage factor Hrp1/Nab4 increased in abundance by 49% consistent with the upregulation of its mRNA and a cellular attempt to normalise 3'ends.

We previously showed that cordycepin-triphosphate (CoTP), rather than cordycepin, mediated the toxicity of the drug in yeast (*Holbein et al., 2009*). To address whether CoTP was also responsible for altered 3' end processing, Δ*ado1* and Δ*adk1* strains were subjected to cordycepin treatment (*Figure 4—figure supplement 2D*). Ado1, converts cordycepin to cordycepin monophosphate (CoMP), whereas Adk1 further converts CoMP to CoTP (*Lecoq et al., 2001*; *Konrad, 1988*). As Δ*ado1* cells were resistant to cordycepin-induced alternative polyadenylation for the gene *OM14* and Δ*adk1* cells were not, this suggested that CoMP, not CoTP, was the mediator of APA. This further separated the roles of cordycepin in RNA incorporation and toxicity from its effect on

alternative polyadenylation. In addition, Δ*nrt1* cells conferred mild resistance to cordycepin-induced APA, suggesting that Nrt1, a high-affinity nicotinamide riboside transporter and low-affinity thiamine transporter (*Enjo et al., 1997*; *Belenky et al., 2008*), acted as a partially redundant transporter for cordycepin (*Figure 4—figure supplement 2D*).

Excess adenine has previously been shown to suppress cordycepin's effect on yeast cell growth (*Holbein et al., 2009*). To determine whether adenine addition was also able to reduce cordycepin-induced APA, wild-type BY4741 cells were treated with cordycepin and adenine. However, adenine co-treatment did not return cells to wild-type cleavage site positions (*Figure 4—figure supplement 2E*). Furthermore, to ask whether other 3' deoxynucleosides can induce APA, cells were treated with the guanosine equivalent of cordycepin, 3' deoxyguanosine (*Figure 4—figure supplement 2E*). However, this analogue caused no changes to cleavage site choice further supporting the idea that cordycepin is unlikely to confer APA through RNA chain termination and indicates that the effect was specific to the adenosine analogue.

## Cordycepin alters yeast intermediate metabolism to promote nucleotide accumulation

The concentration of cordycepin used in our analyses did not significantly impair cell growth (*Figure 4—figure supplement 2F*). To further investigate the impact of the drug treatment, we analysed cellular metabolism. When grown in glucose-rich conditions, yeast preferentially ferments glucose to ethanol rather than perform aerobic respiration despite the presence of oxygen (*Broach, 2012*). To ask whether this was altered by cordycepin treatment, we first performed the Seahorse Energy Phenotyping assay on wild-type BY4741 yeast cells that were treated with cordycepin. This result was then compared with both DMSO treated and untreated cells. All three conditions responded to metabolic stress by increasing oxygen consumption, suggesting that cordycepin did not affect oxygen utilisation (*Figure 5—figure supplement 1A*). However, cordycepin-treated cells did not as efficiently undergo the increase in glycolytic output observed with both control groups (*Figure 5—figure supplement 1B*). In yeast, an increase in extracellular acidification is expected to report carbon dioxide production rather than lactic acid synthesis and indicates a decrease in overall ethanol synthesis in cordycepin-treated cells. As there was sufficient glucose in the media to support an increase in the control-treated cells, this suggests that metabolites of the glycolytic pathway of cordycepin-treated cells were diverted towards alternative pathways.

To probe the extent of metabolic rewiring by cordycepin, a metabolomic analysis was performed on cordycepin-treated cells. Several metabolites involved in glycolytically linked pathways were significantly upregulated following cordycepin treatment (*Figure 5*). The reserve carbohydrate, and stress protectant, trehalose was increased 3.8-fold, which correlated with drug-induced stress (*Jain and Roy, 2009*; *Eleutherio et al., 2015*) and signifies amplified production of storage carbohydrates. This was accompanied by a 3.1-fold increase in acetyl CoA concentration and high levels of this metabolite are associated with increased acetylation of histones (*Cai et al., 2011*; *Donohoe et al., 2012*; *Lee et al., 2014*) and therefore global transcriptional activity. Notably, we observed increased levels of key intermediates involved in nucleotide biosynthesis, including pentose phosphate pathway intermediates (sedoheptulose-7-phosphate and putative ribulose-5-phosphate and/or xylulose-5-phosphate), inosine monophosphate (IMP), adenylosuccinate, dihydroorotate, orotate monophosphate and several nucleotides and deoxynucleotides (CDP, CTP, UDP, UTP, dCDP, dCTP, dADP, and cAMP) indicating a major impact on nucleotide metabolism. Defective APA following cordycepin treatment was thus accompanied by a striking upsurge in nucleotide abundance, which is an important parameter for RNAP II transcription elongation rate in vitro (*Uptain et al., 1997*).

## Slowing transcription attenuates cordycepin's effect on alternative polyadenylation

We and others have previously shown that the use of distal polyadenylation sites is linked to upregulation of transcriptional initiation (*Swaminathan and Beilharz, 2016*) and elongation rate (*Pinto et al., 2011*). To test the idea that a change in nucleotide metabolism caused by cordycepin treatment leads to an increase in transcription rate and thus use of distal poly(A) sites, 4-thiouracil (4tU) labelling was used. This method monitors newly synthesised transcripts through the

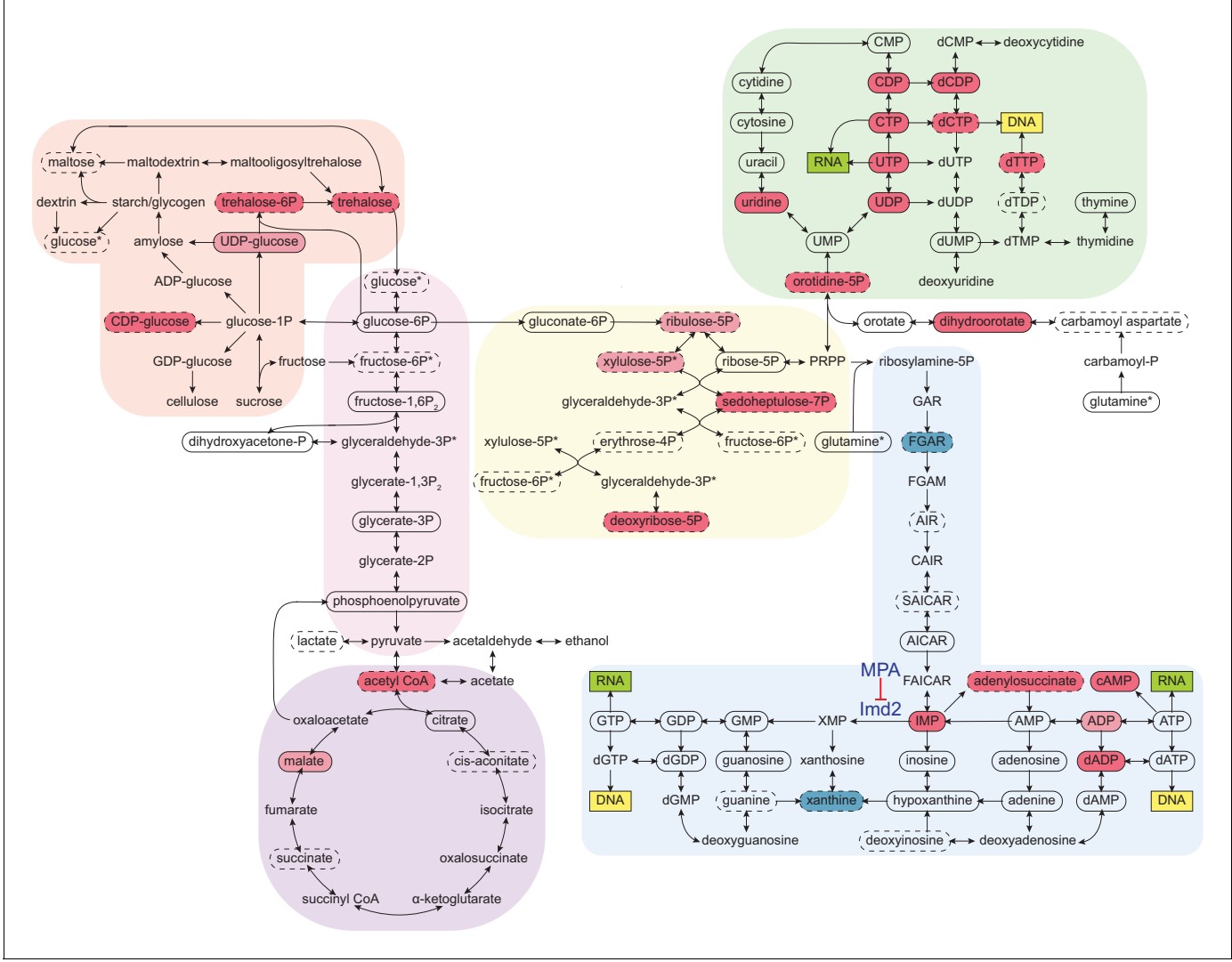

**Figure 5.** Cordycepin alters yeast core metabolism to promote nucleotide biosynthesis. Yeast metabolomics pathway map showing metabolite fold changes following 40 min treatment with 20 μg/ml cordycepin. Solid circles indicate detected confirmed metabolites and dashed circles detected putative metabolites. Coloured circles show metabolites with significant fold change after cordycepin treatment with red indicating an increase in abundance and blue a decrease in abundance. Darker colour is used for metabolites with greater than 50% fold change. Pathways are based on the KEGG database and include starch and sucrose metabolism (orange), glycolysis/gluconeogenesis (pink), citrate cycle (TCA cycle) (purple), pentose phosphate pathway (yellow), pyrimidine metabolism (green), and purine metabolism (blue). Asterix (*) indicates metabolites that appear multiple times on the map. The pharmacological inhibition of Imd2 by mycophenolic acid (MPA) is indicated. Data is representative of six biological replicates and is available in the *Figure 5—source data 1*.

The online version of this article includes the following source data and figure supplement(s) for figure 5:

**Source data 1.** Cordycepin metabolomics data.

**Figure supplement 1.** Glycolytic output increase following mitochondrial stress is suppressed upon cordycepin treatment.

incorporation of 4tU, which subsequently appears as a T to C conversion when sequenced. Wild-type BY4741 cells were treated with cordycepin or DMSO and 4tU for 40 min. mRNA was then analysed using a modified version of the mPAT approach that used an anchored oligo-dT (TV12VN) primer for reverse transcription to focus on APA. For the gene *OM14*, three major cleavage sites were detected (*Figure 6A*).

Treating cells with cordycepin caused increased use of the most distal of these alternate cleavage sites. In 4tU-labelled samples, control cells had all three isoforms newly synthesised, whereas in cordycepin-treated cells, the majority of new transcripts were cleaved at the most distal cleavage site. Therefore, the increased use of longer 3' UTR isoforms was not a result of changed stability between

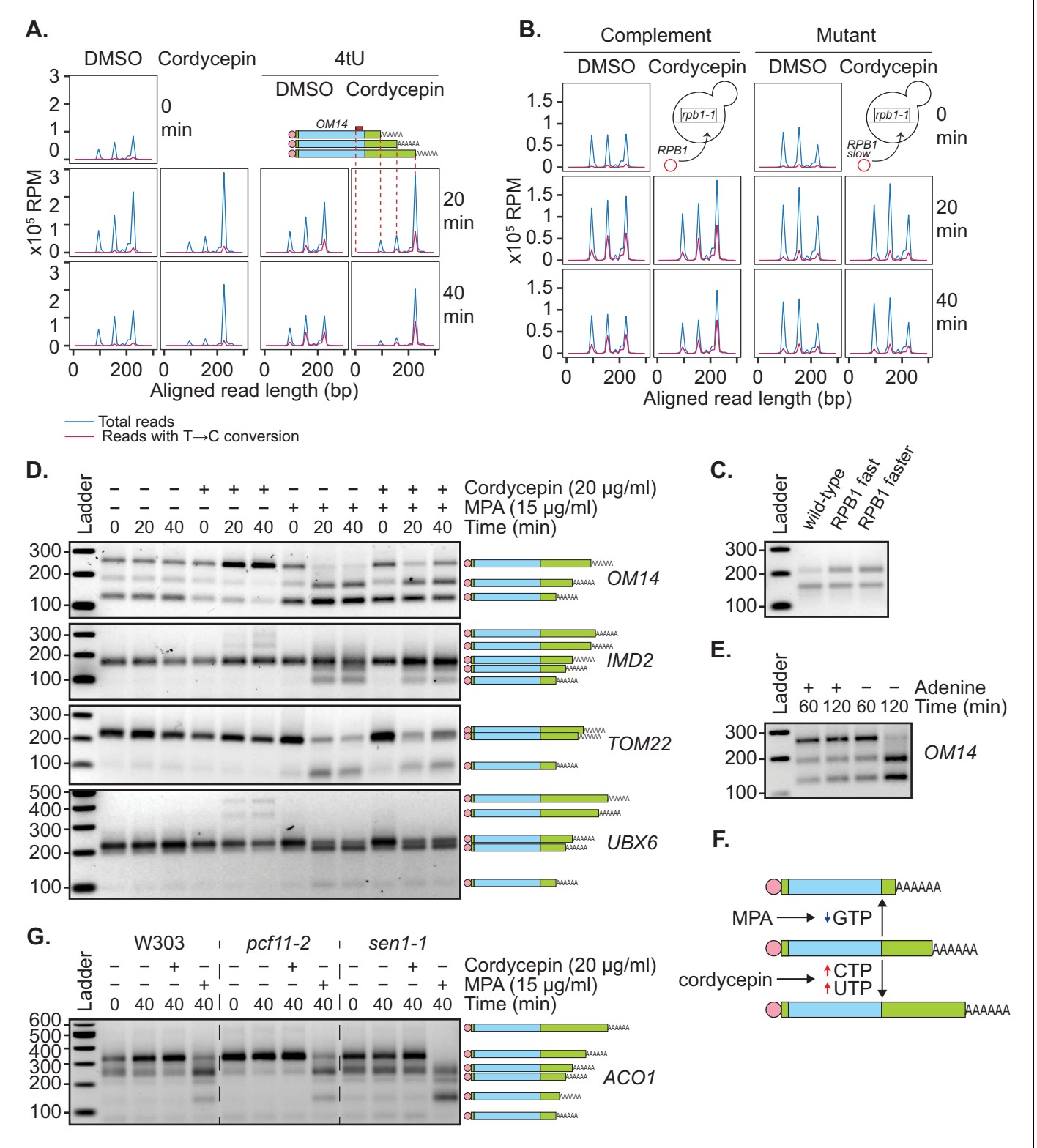

**Figure 6.** Genetic or pharmacological slowing of transcription attenuates cordycepin's effect on alternative polyadenylation. (**A and B**) Yeast cells treated with 20 µg/ml cordycepin or DMSO (control) and 4tU labelled for 20 or 40 min. mRNA was then analysed using the mPAT approach. Reads with T to C conversions represent transcripts synthesised during the 4tU labelling and thus newly produced mRNA. The 3' ends of reads aligned to the gene *OM14*, with the forward primer at 0, are shown in x10⁵ reads per million. This was done for A. Wild-type BY4741 cells and B. *rpb1-1* mutant cells complemented with *RPB1* or an *rpb1* slow mutation. Data is representative of two biological replicates. (**C**) TVN-PAT analysis of the gene OPI3 for cells

*Figure 6 continued on next page*

*Figure 6 continued*

containing wild-type, L1101S 'fast' or E1103G 'faster' Rpb1. Samples were run relative to a 100 bp ladder. Bands correspond to the different APA isoforms of the gene with slower migrating bands indicating a more distal cleavage site and faster bands a more proximal cleavage. (D) TVN-PAT analysis of the genes *OM14*, *IMD2*, *TOM22*, and *UBX6* for yeast cells treated with 20 µg/ml cordycepin and 15 µg/ml MPA for 20 or 40 min. (E) TVN-PAT analysis of the gene *OM14* for wild-type W303 cells (adenine auxotroph) grown with or without adenine hemisulphate (100 µg/ml) for 60 or 120 min. (F) Schematic illustration of the pharmacological induction of APA. MPA treatment decreases GTP availability resulting in shorter 3'UTRs, whereas cordycepin treatment increases CTP and UTP concentrations and promotes 3'UTR lengthening. (G) TVN-PAT analysis of the gene *ACO1* for wild-type W303 and temperature-sensitive mutants *pcf11-2* and *sen1-1*. Cells were grown at 25°C then switched to the semi-restrictive temperature 30°C and treated with 20 µg/ml cordycepin or 15 µg/ml MPA for 40 min.

the different isoforms but was in fact due to the distal cleavage site being preferentially used in new transcripts. However, the total incorporation of 4tU did not significantly change between DMSO and cordycepin samples (mean incorporation of 1.011% in DMSO samples and 0.779% in cordycepin samples, p-value=0.227, 95% CI = −1.329 to 0.866). We argue that this was due to dilution of the label by the de novo nucleotide biosynthesis induced by cordycepin. As a result, an increased transcriptional rate by cordycepin could not be confirmed by an increased rate of 4tU incorporation.

To overcome this, 4tU labelling was repeated using a slow mutant of RNAP II (*Kaplan et al., 2012*). We reasoned that if the change in cleavage site use was caused by cordycepin increasing the transcription rate, a slow polymerase should be able to override this effect. When the *rpb1-1* mutant was complemented with a plasmid-borne copy of wild-type *RPB1* (*Kaplan et al., 2012*), cordycepin caused a shift toward the usage of distal cleavage sites (*Figure 6B*). However, *rpb1-1* complemented with a slow allele of *rpb1* (H1085Y mutation [*Kaplan et al., 2012*]), did not display this change. Therefore, slowing the global elongation rate ablated cordycepin's effect on 3' end processing. In contrast, cells containing fast *rpb1* alleles (*Kaplan et al., 2012*) displayed an increased use of the distal cleavage site for the gene *OPI3* (*Figure 6C*, *Geisberg et al., 2020*) similarly to cordycepin treatment. This observation was consistent with the idea that cordycepin did indeed promote alternative polyadenylation by altering the rate of transcription.

Mycophenolic acid (MPA) is an inhibitor of IMP dehydrogenase, the rate-limiting enzyme in de novo GTP synthesis. MPA treatment consequently interferes with transcription elongation due to substrate limitation (*Mason and Struhl, 2005*). To further demonstrate that cordycepin's APA effect was mediated by changes to nucleotide levels that in turn affected the transcriptional rate, cells were treated with 15 µg/ml MPA. Consistent with the idea that APA hinges on nucleotide levels, we found that MPA indeed caused increased usage of proximal cleavage sites (*Figure 6D*). This was similar to results seen upon elimination of adenine in W303 cells (*Figure 6E*), which require adenine supplementation for growth. Moreover, co-treatment of 20 µg/ml cordycepin with MPA led to a suppression of this shortening (*Figure 6D*). Collectively, these observations suggested that altered RNAP II elongation rate, as a consequence of nucleotide availability, was able to promote APA (*Figure 6F*).

To examine the interaction of nucleotide and cleavage factor levels in cleavage site selection, *pcf11-2* and *sen1-1* cells were treated with cordycepin or MPA (*Figure 6G*). Cordycepin treatment caused no further lengthening of the *ACO1* transcript in the 3'UTR lengthening mutant *pcf11-2* and was unable to suppress the shortening seen in the *sen1-1* mutant. In contrast, treatment with MPA prevented *pcf11-2* 3'UTR lengthening and caused increased shortening in *sen1-1*. This shows that 3'ends cannot extend past their most distal poly(A) site and produce a functional transcript and demonstrates that a reduced nucleotide availability has a stronger impact on cleavage site choice than the *pcf11-2* and *sen1-1* mutations.

## The ability of a gene to undergo alternative polyadenylation is dependent on the local nucleosome landscape

A large proportion of the yeast genome displayed changes in alternative polyadenylation following cordycepin treatment. But why did specific genes alter their 3' ends in response to both loss of the core cleavage machinery and the drug cordycepin whilst others did not? To address this question, we assessed the structural differences between genes that responded to cordycepin treatment and those that maintained their original cleavage sites.

For this analysis, genes were considered to be alternatively polyadenylated if more than 40% of reads switched to another cleavage site following cordycepin treatment and to not undergo alternative polyadenylation if the same cleavage site was primarily used in treated and untreated cells. This was a more stringent classification of APA than was utilised in the end-shift test in *Figure 3B* above. On average, genes with a high level of APA were significantly longer than those without APA (p-value=0.031) with median gene coding region lengths of 1420 bp and 1275 bp, respectively (*Figure 7A*). This suggested that the length of a gene may influence its ability to undergo APA.

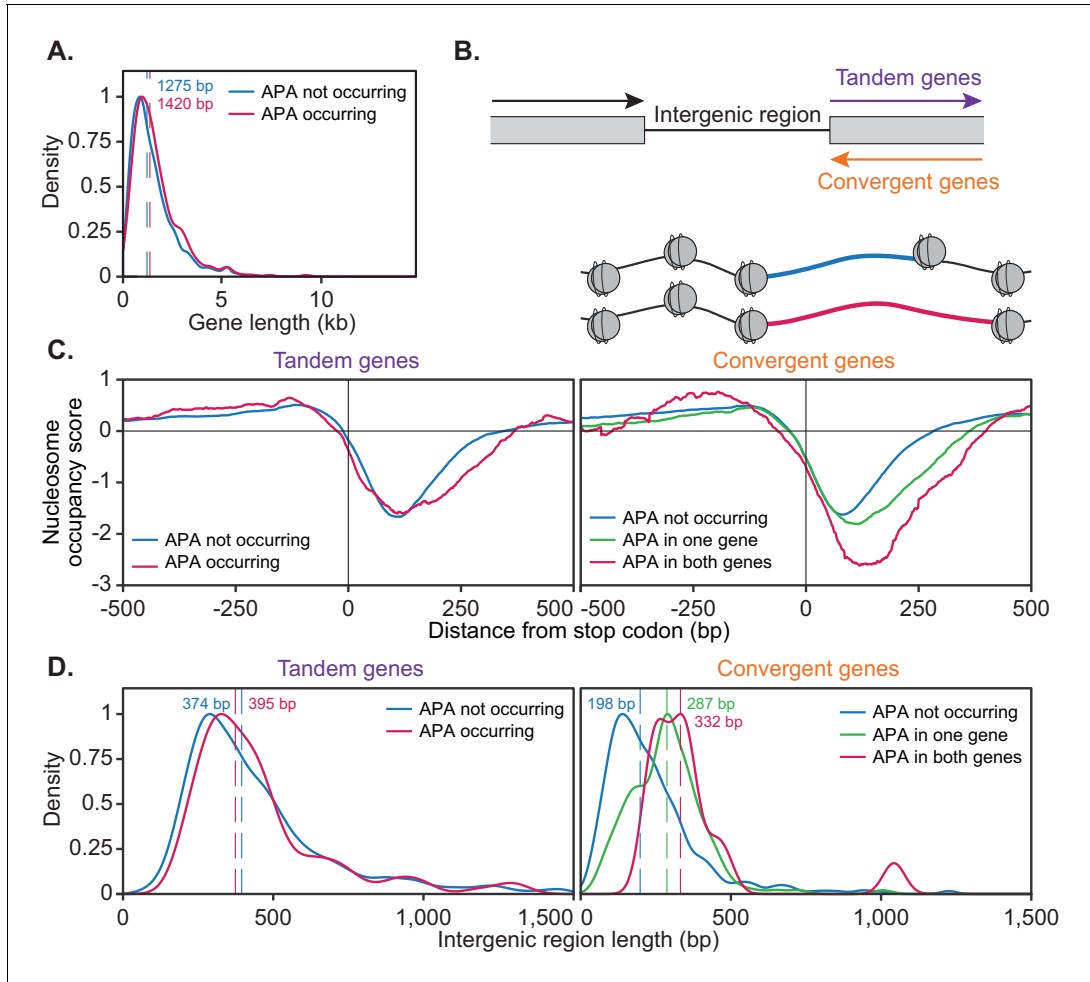

**Figure 7.** The ability of a gene to undergo alternative polyadenylation is dependent on the local nucleosome landscape. Comparison of native genetic features in genes split based on APA following cordycepin treatment. APA was considered to occur if the major cleavage site (greater than 40% of reads) shifted to another site following 20 µg/ml cordycepin treatment for 40 min and to not occur if the major cleavage site remains the same. Genes used in each comparison are included in the *Figure 7—source data 1*. (A) Scaled density plot comparing gene length (in kilobase pairs) for genes in which APA occurs (pink, n = 378) and does not occur (blue, n = 4302). Dashed lines and coloured values indicate the median gene length. T-test p-value=0.031. (B) Schematic of the two types of 3′ end intergenic regions. The downstream gene can either be transcribed in the same direction as the upstream gene (tandem genes) or the opposite direction (convergent genes), in which case two genes will be assigned the same 3′ end intergenic region. (C) Normalised nucleosome occupancy per base pair around the translation end site, averaged across all yeast genes, derived from *Kaplan et al., 2009*. For tandem genes, APA occurred in 128 genes (pink) and did not occur in 1719 genes (blue). T-test p-value=0.055. For convergent genes, APA occurred in both genes for 15 gene pairs (pink), in one gene for 155 pairs (green) and in neither gene for 746 pairs (blue). ANOVA p-value<0.001. Nucleosome occupancy values above zero indicate nucleosome enrichment compared to the genome-wide average. (D) Scaled density plot comparing 3′ end intergenic region length (in base pairs) based on APA up to 1500 bp. Dashed lines and coloured values indicate the median intergenic region length. Tandem gene t-test p-value=0.646, convergent gene ANOVA p-value<0.001.

The online version of this article includes the following source data for figure 7:

**Source data 1.** Genes undergoing APA following cordycepin treatment.

Genes were then further classified based on the directionality of their downstream neighbouring gene (*Figure 7B*). If the neighbouring gene was transcribed in the same direction, the gene of interest was classified as 'tandem', whereas if the adjacent gene was transcribed in the opposite direction, this gene was termed 'convergent'. Convergent genes share a 3' end intergenic region and are referred to as 'gene pairs'. Gene pairs were divided based on whether both, one or neither genes undergo APA.

We next sought to determine whether the chromatin landscape of a gene correlated with alternative polyadenylation. Nucleosomes are positioned at defined locations across the genome. In particular, nucleosome depleted regions are associated with transcription start and stop sites (*Shivaswamy et al., 2008*; *Lee et al., 2004*; *Xu et al., 2009*; *Mavrich et al., 2008*). Focusing on the 3' end, we compared the native nucleosome occupancy of genes based on their responsiveness to cordycepin treatment (*Figure 7C*). This was done with a publicly available dataset of yeast nucleosome occupancy (*Kaplan et al., 2009*).

The nucleosome occupancy landscape of tandem genes did not significantly differ based on APA (p-value=0.055). However, convergent genes showed significant differences in nucleosome occupancy scores between gene pairs in which both, one and neither genes undergo APA (p-value<0.001). On average, the width of the nucleosome depleted region after the stop codon was wider when alternative polyadenylation did occur. This was enhanced when both genes in a pair displayed APA. As these cells were not treated with cordycepin, it appeared that responsive convergent genes had an innately wider nucleosome depleted region at the 3' end of the gene. Therefore, a gene's alternative polyadenylation potential may be linked to a pre-determined framework of nucleosome positioning and the directionality of their neighbouring downstream gene.

In addition to a wider nucleosome depleted region at the 3' end of the gene, cordycepin responsive convergent genes had on average a longer intergenic region between neighbouring genes (*Figure 7D*; median 332 bp for APA occurring in both genes and 198 bp for APA not occurring in either gene, p-value<0.001). Where one gene in a pair undergoes APA and the other did not the intergenic region length was intermediate (median 287 bp) compared to those in which APA occurs in both or neither gene. Increased intergenic region length may provide more room for additional cleavage sites to exist for a single gene and permit transcriptional extension without collision of transcriptional machinery between neighbouring genes. The length of the intergenic region for tandem genes did not significantly correlate with APA (median 397 bp for APA occurring and 380 bp for APA not occurring, p-value=0.856). It, therefore, appears that 3' end nucleosome positioning and the length of the 3' end intergenic region were determinants of APA potential for convergent but not tandem genes.

Based on the data presented here we propose the following model for alternative polyadenylation regulation in *S. cerevisiae* (*Figure 8*). Cleavage site choice exists as a balance between the availability of the cleavage and polyadenylation machinery, it's affinity towards the sequence elements present on the pre-mRNA substrate and the transcriptional elongation rate. When this balance is tipped, it alters which cleavage site is preferentially utilised. High concentrations of the 3' end processing machinery ensure cleavage sites can be efficiently recognised promoting cleavage at more proximal cleavage sites due to their temporal advantage. Conversely, decreased concentrations, or cleavage factor mutations, will delay processing at proximal sites and increase distal cleavage site usage.

Changing the rate of transcription has a similar impact on cleavage site choice as this shifts the ratio between transcript elongation and the availability of the 3' end processing machinery. Faster transcription, as instigated by cordycepin treatment and facilitated by a permissive chromatin landscape, provides less time for the proximal site to be recognised by the cleavage factors before the distal form is also transcribed. The physical coupling of 3' end formation and transcription is likely to promote the preferential usage of the longer 3' UTR isoform once the distal cleavage site is available.

## Discussion

Despite its pervasive nature, the mechanisms of alternative polyadenylation remain poorly understood. The starting point of this study was a systematic analysis of 3' end formation factors, which uncovered that mutant alleles of most of the factors promoted global switching towards distal sites.

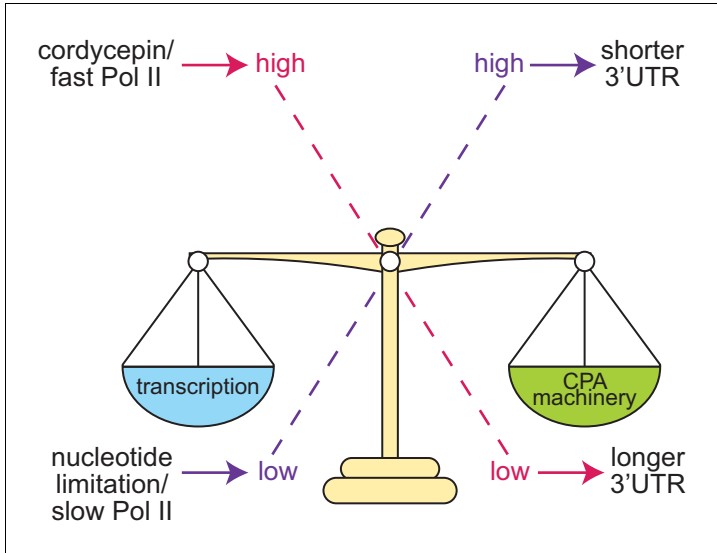

**Figure 8.** Proposed model of cleavage site selection in yeast. Schematic of cleavage site selection in yeast. We propose a kinetic model where balance exists between nucleotide levels, transcriptional rate, and the concentration of the cleavage and polyadenylation machinery. Tipping this scale by alteration of nucleotide concentration or mutation of cleavage factors and RNA polymerase II causes changes to cleavage site choice.

In contrast, a subset of genes shifted toward more proximal sites when the RNA helicase Sen1 was mutated. In addition, we demonstrate that cordycepin treatment caused a global shift to distal poly (A) sites, an effect that we found to be accompanied by an accumulation of nucleotides and their biosynthetic intermediates. In agreement with the idea that changes in cellular nucleotide concentrations impacted the usage of alternative poly(A) sites following cordycepin treatment, we show that mycophenolic acid, which is known to reduce GTP levels, caused a shift to more proximal sites and combined treatment reversed the shifts observed with cordycepin alone. Furthermore, suppression of cordycepin effects by a slow *rpb1* mutant of RNAP II functionally linked transcription elongation to alternative polyadenylation. Since transcription elongation is sensitive to nucleotide levels, we propose that nucleotide metabolism plays a key role in transmitting kinetic coordination of RNAP II elongation and coupled pre-mRNA 3' end processing in the regulation of alternative polyadenylation.

Employing global PAT-seq and targeted mPAT approaches we found that preferred poly(A) sites in wild-type cells shifted towards more distal sites in the majority of strains carrying mutations in subunits of the canonical cleavage and polyadenylation machinery. Overall, our results indicate that gene specific roles for individual 3' end factor subunits or their expression levels play a minor part in the regulation of poly(A) site recognition for the majority of genes. Mutations in these factors can compromise efficient recognition of the more proximal sites, which are then skipped in favour of a stronger more distal site that is more readily recognised by the impaired 3' end processing machinery (*Shepard et al., 2011*; *Beaudoing et al., 2000*). However, as some genes are impacted upon mutation of individual subunits only, there also appears to be a degree of subunit specificity for alternative polyadenylation and some phenotypic variability likely can also be attributed to the specific alleles analysed in this study. Interestingly, changes in APA did not correlate with defective in vitro cleavage activity associated with mutant alleles (*Figure 2B*). A likely explanation for this is that reduced cleavage activity impacted the usage of alternative sites equally and thus there was no apparent shift in the cleavage site. In contrast, in vitro polyadenylation activity did correlate with defective APA in all analysed cases. The reasons for this remain unclear but the mutants analysed with defective in vitro polyadenylation also display RNA-binding activity, which may be compromised giving rise to the defect in APA. The requirement for a functional CTD-binding domain within Pcf11 highlighted the importance of co-transcriptional interactions for normal 3'end formation. Quantitative variations in the strength of 3' end shifts observed for different mutant strains thus likely

reflected distinct functional roles and deficiencies, which can impact on RNA binding, subcomplex interactions and complex stability, and coupling of 3′ end formation to RNAP II transcription.

We observed a unique requirement for the Sen1 helicase for poly(A) site-choice on a subset of 676 genes (*Figure 3A*). Sen1 function has been well characterised in association with the NNS complex where it acts to promote transcription termination of non-coding RNA and of a small number of protein coding genes (*Steinmetz et al., 2006*). The role played by Sen1 in alternative polyadenylation described here is distinct from the previously described roles in RNA processing because the observed *sen1-1* mutant phenotype was not duplicated by *nrd1-5* and *nab3-11* mutations, and thus, Sen1 appeared to act independently of the RNA-binding proteins Nrd1 and Nab3. In contrast, deficiency of the Nrd1-like protein Seb1 in *Schizosaccharomyces pombe* resulted in widespread 3′ UTR lengthening demonstrating that Nrd1's role in mRNA 3′ end formation has not been conserved between budding and fission yeast (*Lemay et al., 2016*). Here, the *sen1-1* allele caused shifting toward sites proximal to those utilised in the wild-type. Therefore, functional Sen1 appeared to promote the usage of downstream poly(A) sites and thus suppressed usage of proximal poly(A) sites rather than promoting termination as was previously seen with non-coding RNA transcription. This may indicate that Sen1 antagonised the function of the core cleavage and polyadenylation machinery, since Pcf11 was inversely required for the usage of more proximal poly(A) sites on more than 80% of the identified Sen1 target genes. Interestingly, this activity was associated with genes that carried an above average length of the open reading frame suggesting that Sen1 may facilitate full-length expression of long genes. The mechanisms underlying Sen1 function in alternative polyadenylation remain unclear and need to be addressed in future studies. We speculated that defective termination of antisense non-coding RNA transcription in *sen1-1* mutants facilitated transcriptional interference that in turn caused an apparent suppression of the use of distal sites on the sense mRNA transcript. Such a role would, however, be expected to involve Nrd1 and Nab3 and therefore may not apply here. Alternatively, since Sen1 does interact with the RNAP II CTD (*Vasiljeva et al., 2008*; *Han et al., 2020*), it may impact on the interaction of other CTD binding factors, like Pcf11 (*Grzechnik et al., 2015*), and in this way interfere with efficient 3′ end formation and/or termination of transcription that could shift poly(A) site usage from proximal to distal positions. Another possibility may be that Sen1 helicase activity was required to resolve RNA structures that form in proximity to the RNAP II exit channel, which may provoke transcriptional pausing and promote preferential usage of proximal sites if unresolved. Interestingly, an NNS-independent requirement for Sen1 has been reported previously in the protection of replication fork integrity that may be associated with pathology of Senataxin deficiencies (*Alzu et al., 2012*) but it remains unclear if this is related to Sen1 function in APA as described here.

Recent analysis of 3′ end factor interaction on nascent pre-mRNA suggested that the overall architecture of complex assembly across poly(A) sites is conserved between yeast and mammals (*Baejen et al., 2014*). In the mammalian system cleavage factor II subunits have been associated with the selection of proximal sites (*Kamieniarz-Gdula and Proudfoot, 2019*; *Ogorodnikov et al., 2018*; *Turner et al., 2020*; *Wang et al., 2019*), while cleavage factor I (CFIm) promotes the usage of distal sites suggesting that CFIm acts to represses cleavage at proximal sites (*Martin et al., 2012*; *Masamha et al., 2014*). As no CFIm homologues are known in yeast the repressive role of the core cleavage and polyadenylation machinery may not be conserved. Instead, the compact nature of the yeast genome with short intergenic regions may have favoured mechanisms that promote the preferential usage of proximal poly(A) site in order to avoid transcriptional interference between neighbouring genes (*Gullerova and Proudfoot, 2010*). The genes that require Sen1 for selection at distal sites may form a distinct group that was characterised by long open reading frames and it will be interesting to see whether the Sen1 homologue Senataxin plays a similar role in humans.

Cordycepin's ability to terminate RNA synthesis following its incorporation into the polynucleotide chain has been widely demonstrated (*Horowitz et al., 1976*; *Müller et al., 1977*). We have previously shown that cordycepin interferes with mRNA 3′ end formation independently of RNA chain termination causing 3′ UTR read-through in the genes *ACT1*, *ASC1* and *CYH2* (*Holbein et al., 2009*) opening the possibility that the drug may impact on alternative polyadenylation. Through PAT-seq analysis we have shown here that low-dose cordycepin treatment indeed stimulated alternative polyadenylation and distal cleavage site usage globally of similar extent to those observed with 3′ end factor mutants. Cordycepin effects were most comparable to shifts in *pcf11-2* and *ysh1-13* yeast strains. However, as no alternative polyadenylation profile of a 3′ end factor matched that of

cordycepin, the drug was unlikely to act through direct interaction with specific proteins. Furthermore, although cordycepin did provoke coordinated changes in 3' end factor RNA expression, only Hrp1/Nab4 showed significant changes in protein levels following a 40 min exposure to the drug. This indicated that changes in protein levels were unlikely to cause the cordycepin induced alternative polyadenylation, which was detected within 10 min of drug treatment. Our metabolomics analysis revealed accumulation of nucleotides and their biosynthetic intermediates in wild-type cells exposed to cordycepin. An increase in the pool of cellular nucleotides may reduce cordycepin toxicity by dilution. This idea is in line with our previous observation that poly(A) polymerase mutants displayed cordycepin resistance due to increased levels of free ATP (*Holbein et al., 2009*).

Importantly, nucleotide concentration is a key determinant of RNAP II processivity and elongation rate (*Mason and Struhl, 2005*; *Uptain et al., 1997*). Consistent with this idea drugs like mycophenolic acid and 6-azauracil that reduce GTP and UTP levels, respectively, act synergistically with defects in trans-acting factors required for efficient elongation causing a reduced rate of nucleotide incorporation, RNAP II pausing and dissociation of RNAP II from chromatin (*Exinger and Lacroute, 1992*; *Shaw and Reines, 2000*; *Mason and Struhl, 2005*). The increased dwelling time of RNAP II on the chromatin template is thought to enhance the probability of associated 3' end formation to take place and it was recently suggested that the rate of transcriptional elongation controls alternative polyadenylation (*Geisberg et al., 2020*) and *Figure 6C*. Consistent with these observations increased nucleotide levels observed following cordycepin treatment are predicted to enhance the rate of RNAP II elongation, which will reduce the window of opportunity for 3' end formation factors to associate with the transcript at proximal sites, and allowing for the transcription and use of distal poly(A) sites. The link between nucleotide levels and alternative polyadenylation is underscored by our observation that mycophenolic acid, which slows RNAP II transcription through reduced GTP levels, promoted usage of proximal sites and suppressed the effect of cordycepin. Similarly, a slow *rpb1* allele mirrored the effects of mycophenolic acid to suppress the effects of cordycepin on alternative polyadenylation. Importantly, distal site usage in *pcf11-2* was not further induced by cordycepin, indicating an upper limit to 3' lengthening, whereas depletion of GTP by MPA served to normalise 3' end choice.

In addition to nucleotide levels and *trans*-acting factors, the structure of the chromatin template is an important parameter for RNAP II elongation. Increased nucleosome occupancy at the 3' end of the gene reduces the rate of transcription due to the stalling of RNAP II (*Hodges et al., 2009*). This provides the 3' end formation machinery with more time to recognise this first site prior to the distal cleavage site being transcribed, meaning this position is nearly exclusively used. A stable nucleosome between potential cleavage positions is therefore predicted to provide a more proximal site with a kinetic advantage. Likewise, when the nucleosome depleted region is wider, the proximal site is expected to have less temporal advantage. Consistent with this idea we showed that genes that switch their poly(A) site in response to cordycepin treatment differ in their chromatin landscape compared to those that retain their original poly(A) sites. Convergent genes that were subject to alternative polyadenylation following cordycepin treatment tended to have wider intergenic regions downstream of the coding sequence, which likely allowed for transcriptional extension without collision of the transcriptional machinery between neighbouring genes. In contrast, no such widening of the intergenic region was observed for genes that are transcribed in tandem consistent with transcriptional interference being no issue at those genes. Moreover, we found that cordycepin-responsive convergent genes had innately wider nucleosome depleted regions at the 3' end of the gene consistent with the idea that nucleosome density restricts the occurrence of alternative polyadenylation.

Taken together, our results support a model where kinetic competition governs the selection of alternative poly(A) sites. On the one hand, proximal sites are transcribed first and thus carry a temporal advantage. The strength of a given poly(A) site and the availability of 3' end factors will determine the kinetics of cleavage and polyadenylation at those sites. Transcription of distal sites, on the other hand, will be achieved only under conditions which delay rapid processing at proximal sites and which favour rapid elongation by RNAP II. Once distal sites are available, however, they will compete with proximal sites for interaction with the 3' end formation machinery. The closer vicinity to elongating RNAP II may then favour the distal site due to the physical interaction of 3' end formation factors and the CTD of RNAP II. A more complete understanding of the mechanisms of alternative polyadenylation will thus require the analysis of a large number of nuclear proteins, which

directly impact on gene transcription by RNAP II, chromatin structure and its coupling to pre-mRNA 3' end formation.

## Materials and methods

### Yeast cell growth conditions

For cleavage factor mutant experiments, yeast strains were grown in YPAD rich media (1% yeast extract, 2% peptone, 2% glucose, 100 µg/ml adenine hemisulphate) at 25°C to an $OD_{600}$ of 0.6 then transferred to 37°C for 1 hr before cells were harvested.

For confirmation of temperature sensitivity, yeast strains were grown overnight in YPAD rich media at 25°C. Ten-fold serial dilutions were spotted on YPAD plates (1% yeast extract, 2% peptone, 2% glucose, 100 µg/ml adenine hemisulphate, 20 mg/ml agar) and grown for 2 days at either 25°C or 37°C.

For cordycepin treatment experiments, BY4741, Δnrt1, Δado1, and Δadk1 yeast strains were grown in synthetic media (0.67% yeast nitrogen base [without amino acids or carbohydrates; with ammonium sulphate], 2% glucose, 20 µg/ml uracil, 20 µg/ml methionine, 20 µg/ml histidine, and 100 µg/ml leucine) at 30°C to an $OD_{600}$ of 0.6. W303, pcf11-2 and sen1-1 cells were grown in synthetic media at 25°C (0.67% yeast nitrogen base [without amino acids or carbohydrates; with ammonium sulphate], 2% glucose, 20 µg/ml uracil, 20 µg/ml tryptophan, 20 µg/ml histidine, 100 µg/ml leucine, 100 µg/ml adenine hemisulphate) to an $OD_{600}$ of 0.6 then moved to the semi-restrictive temperature 30°C. 20 µg/ml cordycepin, or an equal volume of DMSO (vehicle control) was then added and incubation was continued for the desired treatment lengths before cells were harvested. Where mycophenolic acid (MPA) was used, it was added at 15 µg/ml to cells grown exactly as for cordycepin treatment. All samples were collected immediately by centrifugation, washed with 1 mL ice cold milliQ $H_2O$ containing sodium azide (0.1% w/v) and pellets frozen at −80°C until RNA extraction.

For confirmation that a non-lethal dose of cordycepin was applied, BY4741 cells were grown overnight in YPAD rich media at 30°C. Ten-fold serial dilutions were spotted on SC plates (0.67% yeast nitrogen base (without amino acids or carbohydrates; with ammonium sulphate), 2% glucose, 20 µg/ml uracil, 20 µg/ml methionine, 20 µg/ml histidine and 100 µg/ml leucine, 20 mg/ml agar) and grown for 2 days at 30°C with 20 µg/ml or 120 µg/ml cordycepin. An equivalent volume of DMSO was used as a control.

For the cordycepin PAT-seq, metabolomics and proteomics datasets, two replicates of $OD_{600}$ 0.6 BY4741 cells treated with 20 µg/ml cordycepin for 0, 5, 10, 20, or 40 min were prepared over 3 separate days (six biological replicates total). Harvested cells were washed with ice cold water containing sodium azide (0.1% w/v) and frozen with liquid nitrogen. Two replicates were used for PAT-seq, six for metabolomics and five for proteomics.

For W303 growth with or without adenine, cells were grown in W303 background synthetic media supplemented with 100 µg/ml adenine hemisulphate at 30°C to an $OD_{600}$ of 0.6. Cells were then resuspended in the same media with or without adenine hemisulphate and grown for 1 or 2 hr.

Rpb1 fast mutants were created by transforming L1101S (fast) and E1103G (faster) Rpb1 mutant alleles (*Kaplan et al., 2012*) on *LEU2* pRS315 based plasmids into CKY283 cells followed by replica plating on 5-fluoroototic acid to remove cells retaining the wild-type *RPB1 URA3* plasmid. Cells were then grown in YPAD media until an $OD_{600}$ of 0.6.

### Total RNA extraction

Total RNA extraction was carried out using the hot phenol method (*Schmitt et al., 1990*) with slight modifications. Cell pellets were resuspended in 400 µl AE buffer (50 mM sodium acetate pH 5.2, 10 mM EDTA) on ice. A total of 33 µl of 25% SDS and 400 µl acid phenol was added then samples were vortexed and transferred to a 65°C water-bath for 20 min with intermittent vortexing. Samples were incubated on ice for 5 min to precipitate the SDS then centrifuged at top speed for 5 min at 4°C in a bench-top centrifuge. The supernatant (both phases) was transferred to a fresh tube and 400 µl of chloroform was added. Samples were vortexed thoroughly and centrifuged at top speed for 5 min at room temperature. The aqueous phase was then transferred to a fresh tube, avoiding the interphase and another 400 µl of chloroform was added. Following vortexing and centrifugation, the aqueous phase was transferred to a fresh tube containing 1/10th volume of 3 M sodium acetate pH

5.2 and an equal volume of isopropanol was added. Tubes were mixed by inversion then centrifuged at top speed for 20 min at 4℃. Supernatant was removed and pellets were washed with 80% ethanol and centrifuged at top speed for a further 10 min. Pellets were air-dried at room temperature and resuspended in RNase-free dH$_2$O.

## PAT-seq

PAT-seq libraries were prepared as described previously (*Harrison et al., 2015*). A 5′ biotinylated PAT-anchor primer, compatible with Illumina index primers, was used to create a 3′ tag on adenylated RNA. This ensures that reverse transcription only occurs from true 3′ ends rather than internal A repeat regions. For this 1 μg total RNA was combined with 1 μl Bio-EE primer and brought to a volume of 12 μl with dH$_2$O in 200 μl PCR tubes. Mixtures were incubated at 80℃ for 5 min then cooled to 37℃ for 5 min. Once cooled, tubes were flash-centrifuged and 8 μl of master mix was added containing 4 μl 5x SuperScript III buffer, 1 μl 100 mM DTT, 1 μl 10 mM dNTPs, 1 μl RNaseOut and 1 μl Klenow polymerase per reaction. Samples were mixed thoroughly by inversion then flash-centrifuged and incubated at 37℃ for 30 min. The temperature was then increased to 80℃ for 5 min and 80 μl digest buffer (300 mM NaCl, 10 mM Tris-HCl pH 7.5) added before incubating for a further 10 min. Samples were then transferred to ice for 5 min.

RNA was then exposed to limited fragmentation by mixing with 100 μl 1:1000 diluted (with digest buffer) solution of RNase T1 for 2 min on ice in phase-lock tubes. This was followed by immediate phenol/chloroform RNA extraction with 200 μl 1:1 acid phenol: chloroform mix and vigorous shaking to stop the reaction. After centrifugation at top speed for 5 min in a benchtop centrifuge, 200 μl of chloroform was added, tubes were shaken and centrifuged again. The top layer was then transferred to new tubes containing 100 μl streptavidin magnetic beads (pre-washed in 55℃ digest buffer) and incubated at 37℃ with shaking for 15 min to allow binding of the DNA-RNA duplex to the beads. Extended 3′ fragments were collected on the streptavidin magnetic beads and resuspended in 200 μl 1/10th strength 5x SuperScript III buffer at 55℃ for 5 min. Fragments were collected on beads again then washed with 100 μl 1x T4 DNA ligase buffer (+ATP) at 37℃ for 5 min with shaking.

5′ phosphorylation of extended 3′ fragments was carried out using T4 Polynucleotide Kinase. 1 μl T4 Polynucleotide kinase and 19 μl T4 DNA ligase buffer (+ATP) was added and samples were incubated at 37℃ for 30 min with shaking. Tubes were then transferred to a 70℃ heat block for 10 min then cooled slowly to room temperature to inactivate the enzyme. 200 μl of 1/10th strength 5x SuperScript III buffer was added and samples were transferred to 200 μl PCR tubes then concentrated using the beads. 7 μl of ligation master mix was added containing 2 μl dH2O, 2 μl stoichiometrically pre-annealed PAT-seq Splint A and Splint B, 3 μl 2 x hybridisation buffer (40% PEG 8000, 1x SuperScript buffer) per reaction. Samples were then incubated at 65℃ for 10 min then 16℃ for 5 min. 8 μl 24% PEG 8000, 2 μl 10 x RNL2 buffer and 2 μl RNA ligase two was added and samples incubated at 16℃ for 16 hr.

Excess 5′ splint was removed by adding 100 μl 1/10th strength 5 x SuperScript buffer and concentrating on the magnetic beads. Reverse transcription was performed using the PAT-seq end-extend primer and SuperScript III on the magnetic matrix. A total of 18 μl master mix was added to each sample containing 11 μl dH$_2$O, 4 μl 5x SuperScript III buffer, 1 μl 100 mM DTT, 1 μl 10 mM dNTPs and 1 μl RNaseOut per reaction. Samples were then incubated at 70℃ for 5 min then cooled to 55℃. One μl SuperScript III was added to each tube, mixed by pipetting and incubated at 55℃ for a further 60 min and heated to 70℃ for 5 min.

The cDNA was size selected by elution from the beads in 2x formamide gel loading buffer and electrophoresis on a 6% urea-PAGE alongside a 25 bp DNA ladder. Fragments selected were between 125 bp and 300 bp in length.

Following gel excision, the cDNA was eluted using the 'crush and soak' method. The excised gel slices were placed inside 0.5 ml pierced lobind tubes in 2 ml lobind tubes. These were centrifuged at 13,000 g for 3 min and 300 μl dH$_2$O added to the bottom tube which was incubated at 4℃ with gentle shaking overnight. Eluted material and gel slurry were transferred to a corning spin-x column and centrifuged at top speed for 3 min. Supernatant was then transferred to a 1.5 ml lobind tube containing 2 μl glycoblue and 35 μl 5 M ammonium acetate and mixed thoroughly. A total of 350 μl isopropanol was added and tubes were incubated on ice for 5 min before being centrifuged at 13,000 g for 20 min at 4℃. Pellets were then washed with 80% ethanol and air-dried before being suspended in 30 μl dH$_2$O. Ten μl of the purified cDNA was used as input for amplification with 1 μl

PAT-seq Universal forward primer, 2 µl TruSeq Index PCR reverse primer, 50 µl 2x Amplitaq Gold 360 Master Mix and 37 µl dH$_2$O. Amplification was performed with cycle conditions of 94°C for 10 min, then 16 cycles of 94°C for 20 s, 60°C for 20 s and 72°C for 30 s, followed by 1 min at 72°C. A different TruSeq primer was used for each experimental condition. PAT-seq libraries were sequenced on the Illumina Hiseq 1500 platform with 100 base rapid chemistry per the manufacturer's instructions at the Monash Health Translation Precinct medical genomic facility in Clayton, Victoria. mPAT mPAT libraries were prepared as previously described (*Beilharz et al., 2017*). To create complementary DNA (cDNA), 1 µg of total RNA and 1 µl of the anchor primer mPAT reverse were combined and brought to a volume of 11 µl with dH$_2$O in 200 µl PCR tubes. Mixtures were incubated at 80°C for 5 min then cooled to 37°C for 1 min. Once cooled, tubes were flash-centrifuged and 8 µl of master mix was added containing 4 µl 5x SuperScript III buffer, 1 µl 100 mM DTT, 1 µl 10 mM dNTPs, 1 µl RNaseOut and 1 µl Klenow polymerase per reaction. Samples were mixed thoroughly by inversion then flash-centrifuged and incubated at 37°C for 30 min. The polymerase was then inactivated by increasing the temperature to 80°C for 10 min before cooling to 55°C for 1 min. At this temperature, 1 µl of SuperScript III was added to each tube and a further 10 min of incubation at 55°C was carried out prior to mixing by inversion and flash-centrifuging. This is done to maintain the temperature during the initial stages of reverse transcription and prevent internal priming. Samples were then incubated at 55°C for 30 min followed by inactivation of reverse transcription at 80°C for 5 min and cooled to 12°C.

For PCR amplification, cDNA was diluted 1:5 with the addition of 100 µl of dH$_2$O. 25 µl diluted cDNA was added to fresh 200 µl PCR tubes with 75 µl master mix containing 1 µl mPAT reverse primer, 1 µl pooled forward primer, 50 µl Amplitaq Gold 360 Master Mix and 23 µl dH$_2$O per reaction. The pooled forward primer contained 46 gene-specific primers that each had a universal 5′ extension for sequential addition of the 5′ (P5) Illumina elements. Amplification was performed with cycle conditions of 95°C for 10 min, then 5 cycles of 95°C for 30 s, 60°C for 30 s and 72°C for 30 s, followed by 1 min at 72°C.

Amplicons were then purified using Macherey-Nagel NucleoSpin columns per manufacturer's instructions and NT1 buffer diluted 1:1 with dH$_2$O. 49 µl of 60°C dH$_2$O was used for the final column elution. A second round of PCR amplification was carried out for each sample using 47 µl purified amplicon, 1 µl P5 universal forward primer, 2 µl Illumina TruSeq indexed primer and 50 µl Amplitaq Gold 360 Master Mix. A different TruSeq primer was used for each experimental condition and amplification as above was repeated for 10 cycles. All PCR reactions were pooled then purified using Nucleospin columns with 1:3 diluted NT1 buffer and a final elution volume of 5 µl dH$_2$O per original sample number. mPAT sequencing was done using MiSeq Reagent kit v2 with 300 cycles (300 bases of sequencing) on the Illumina MiSeq platform according to the manufacturer's specifications at Micromon Genomics, Monash University, Australia.

## Sequencing data processing and statistical tests

PAT-seq and mPAT reads were processed using the Tail Tools pipeline (*Harrison et al., 2015*). Read counts were produced at the level of whole genes and at the level of peaks detected by the pipeline. The reference annotation used was Ensembl *Saccharomyces cerevisiae* version 82 for the cordycepin PAT-seq, version 89 for the mPAT, cordycepin mTVN-PAT and NNS PAT-seq and version 93 for the *rpb1* mTVN-PAT and cleavage factor PAT-seq. These counts were then used to test for differential gene expression and APA. Raw sequencing and processed data has been uploaded to GEO, accession number GSE160539.

Differential gene expression was tested for between experimental groups, using the Fitnoise library (https://github.com/pfh/fitnoise; *Harrison, 2018*) as part of the Tail Tools pipeline, after log transforming and weighting peak counts with voom (*Law et al., 2014*) from the limma R package (RRID:SCR_010943) after TMM library size normalisation (*Robinson and Oshlack, 2010*). Fitnoise is an implementation of Empirical Bayes moderated t-tests on weighted linear models as described in *Smyth, 2004*. Genes for which no relevant sample had at least 10 reads were removed before testing.

Shifts in APA usage was analysed as previously described (*Turner et al., 2020*). Peaks were first assigned to genes with up to 400 bp down-strand of the gene (but not proceeding into another gene on the same strand) were counted as belonging to that gene. Next, peak counts were log$_2$ transformed and linear models were fitted for each peak, using TMM normalisation, voom, and

limma, producing as coefficients estimates of the $\log_2$ abundance of reads at each peak in each of two experimental conditions. For each gene, a 'shift score' was then estimated using these fitted coefficients, as follows.

For PAT-seq experiments, considering a single gene, for two conditions $i = A, B$, and n peaks $j = 1...n$, there is a fitted coefficient $\beta_{i,j}$, and the proportion of reads for each peak within a condition is

$$p_{i,j} = \frac{2^{\beta_{i,j}}}{\sum_{k=1}^{n} 2^{\beta_{i,k}}}$$

A shift score $s$ is then calculated, ranging from $-1$ to 1 where $-1$ indicates all reads in condition B are up-strand of condition A and one indicates all reads in condition B are down-strand of condition A.

$$s = \sum_{i=1}^{n} \sum_{j=1}^{n} \mathrm{sgn}(j - i) p_{A,i} p_{B,j}$$

where $\mathrm{sgn}$ is the sign function. In the simplest case of two peaks, $s$ is equal to the proportion distal peak usage in B minus the proportion distal peak usage in A.

The fitted coefficients $\beta_{i,j}$ have an associated error covariance matrix, which can be propagated through these steps by the delta method to obtain an approximate standard error associated with $s$. Shift scores and associated standard errors are calculated for each gene with two or more peaks, then the topconfects R package (*Harrison et al., 2019*) is used to provide 'confect' inner confidence bounds on the shift scores with correction for multiple testing. Confect values are only given for significantly non-zero shifts, with a False Discovery Rate of 0.05. Ordering results by absolute confect values and reading down this list only as far as is desired, a False Coverage-statement Rate of 0.05 is guaranteed for these confidence bounds.

For mPAT experiments, we use the same $s$ statistic as above, but with a simplified calculation method. Comparing two samples, A and B, with $c_{A,j}$ and $c_{B,j}$ read count at each peak $j$ respectively we calculate $p_{i,j}$ as $p_{i,j} = \frac{c_{i,j}}{\sum_{k=1}^{n} c_{i,k}}$ and calculate $s$ as above.

## Metabolomics

Cell pellets obtained from 10 ml culture at $OD_{600}$ of 0.6 were resuspended in equal volume (10 µl) H2O and cells broken open with cryo-pulverisation using a Retsch CryoMill (50 small ball bearings per sample, $4 \times 45$ s at a frequency of 30 Hz with intermittent cooling in liquid nitrogen for 1 min). 140 µl CMW extraction solvent (1:3:0.8 chloroform:methanol:H2O v/v/v containing the internal standards 1 µM CHAPS, 1 µM CAPS, 1 µM PIPES and 1 µM TRIS) was added and samples vortexed intermittently with ball bearings for 5 min (20 s vortexing, 40 s on ice). Supernatant and cell debris were moved to a fresh tube and intermittent vortexing continued for 15 min. Samples were then centrifuged at top speed for 10 min at 4°C. Supernatant was extracted and stored at −80°C until analysis by liquid chromatography-mass spectrometry.

Chromatography was performed using a Dionex Ultimate 3000 UHPLC system (Thermo) and a ZIC-pHILIC column (Merck) with ammonium carbonate and acetonitrile in the mobile phase (*Creek et al., 2013*) coupled to a Q-Exactive high-resolution mass spectrometer (Thermo). The instrument was operated in both positive and negative ion mode. Parameters for the high-performance liquid chromatography and mass spectrometry analysis were as previously described (*Stoessel et al., 2016*).

Untargeted metabolomics data analysis was performed using the freely availably software packages mzMatch (*Scheltema et al., 2011*) and IDEOM (*Creek et al., 2012*) as previously described (*Trochine et al., 2014*; *Stoessel et al., 2016*). A total of 226 authentic metabolites were used for verification of retention times and to aid metabolite identification. Identification of metabolites with these standards have high confidence (MSI level 1) and are highlighted yellow in *Figure 5—source data 1*. Other metabolites were putatively identified (MSI level 2–3) using exact mass and predicted retention times from KEGG, Lipidmaps and MetaCyc databases (*Creek et al., 2011*). Mean peak height was used for relative quantification and unpaired Welch's T test used for statistical analysis. Data was not normalised, with analysis of the four spiked internal standards (CHAPS, CAPS, PIPES,

and TRIS), total ion current chromatograms, median peak heights and pooled quality control samples analysed throughout the batch used to determine signal reproducibility. It is noted that one replicate from each time point was excluded during analysis due to the observation of unwanted phase separation in the samples (likely due to excess residual water in the pellet).

## Gene and intergenic region length and nucleosome occupancy analysis

Alternative polyadenylation was based on the cordycepin PAT-seq experiment. Reads per million for each peak and sample were calculated using edgeR's TMM library size correction. A primary peak was assigned for each gene for untreated BY4741 cells (t0) and cells treated with 20 µg/ml cordycepin for 40 min (t40). The primary peak must have at least RPM $\geq 5$ and account for at least 40% of reads for that gene at that time point. Where no primary peak was assigned, the gene was discarded. 4410 genes had a primary peak at both t0 and t40. If the primary peak differed at t0 and t40, the gene was considered to undergo APA whereas if the primary peak remained the same after cordycepin treatment, no APA has occurred.

The reference annotation used was Ensembl *Saccharomyces cerevisiae* version 82. Gene and intergenic region width were plotted as scaled density plots so that density peaks for APA and non-APA genes were comparable despite different gene numbers. For gene width, genes were split based on only APA with 378 genes undergoing APA and 4032 genes not undergoing APA. To test whether there was a significant difference between gene length in APA vs. non-APA genes an unpaired two-sided t-test was used. This returned a value of p=0.031 indicating that the difference was significant.

Using the GenomicRanges package (*Lawrence et al., 2013*, RRID:SCR_000025), intergenic regions annotations were created by finding the difference between unannotated chromosomes and the ENSEMBL gene annotations. This was not strand-specific so that intergenic regions are based on genes being on either strand. These were then assigned to the upstream gene so that each allocated intergenic region is at the 3' end of its gene.

Intergenic regions were further characterised into tandem and convergent based on the directionality of their 3' end neighbour gene. The gene on the other side of the intergenic region was classified as either going in the same direction as the primary gene ('tandem') or in the opposite direction ('convergent'). Tandem gene intergenic regions are only assigned to one primary gene as they are on the 3' end of one gene and the 5' end of the neighbouring gene, whereas convergent gene intergenic regions are on the 3' end of two genes and are therefore assigned to two genes, or a 'gene pair'. 1847 genes were characterised as tandem and 2405 genes as convergent.

For tandem genes, 128 genes undergo APA and 1719 genes do not. When limited to an intergenic region width of 1500 bp (due to outliers of up to 4670 bp), 125 genes undergo APA and 1677 do not. To test whether there was a significant difference between tandem gene intergenic region length in APA vs. non-APA genes an unpaired two-sided t-test was used. This returned a value of p=0.856 when considering all genes and p=0.646 when including only intergenic regions up to 1500 bp indicating that the difference was not significant.

Convergent gene pairs were split based on whether both genes undergo APA, one gene undergoes APA or neither gene undergoes APA. Pairs for which APA information was not available for both genes were discarded. APA occurred in both genes for 15 gene pairs (30 genes), in one gene for 155 gene pairs (310 genes) and in neither gene for 746 gene pairs (1492 genes). To test whether there was a significant difference between convergent gene intergenic region length based on APA, analysis of variance (ANOVA) was used. This returned a value of p<0.001 indicating that the difference was significant.

Nucleosome positioning data was obtained from *Kaplan et al., 2009*. Yeast nucleosome DNA was prepared from log-phase cells grown in YPD treated with micrococcal nuclease (MNase). $\text{Log}_2$ normalised nucleosome occupancy per base pair values were used. Genes were characterised as above as tandem or convergent and then split based on APA. Nucleosome positioning data for 500 bp either side of the translational end site for each gene was then averaged for each category. A nucleosome occupancy value of 0 represents the genome-wide average. Values above zero indicate nucleosome enrichment relative to the genome-wide average and values below zero a relatively nucleosome depleted region. To test whether there was significant difference in 3' end nucleosome occupancy based on APA, an unpaired two-sided t-test was used for tandem genes and ANOVA

was used for convergent genes. These returned values of p=0.055 and p<0.001, respectively indicating that the difference was significant convergent genes but not tandem genes.

## 4tu labelling

BY4741 cells were grown in synthetic media (0.67% yeast nitrogen base without amino acids, 2% glucose, 20 µg/ml uracil, leucine 100 µg/ml, histidine 20 µg/ml, methionine 20 µg/ml) at 30°C. The Pol II mutant *rpb1-1* cells (*Olivas and Parker, 2000*), carrying plasmids either with either *RPB1* wild type (pCK859) or the *rpb1* H1085Y slow mutation (pCK870) (*Kaplan et al., 2012*; *Malik et al., 2017*) were grown in synthetic media (0.67% yeast nitrogen base without amino acids, 2% glucose, 20 µg/ml uracil) at 30°C.

4tU labelling was carried out according to the timelapse-seq method (*Schofield et al., 2018*) with modifications for the yeast model system. This involves the use of 4-thiouracil (4tU) rather than 4-thiouradine (4sU), which, is readily taken up by yeast cells (*Munchel et al., 2011*) without expressing a nucleoside transporter (*Miller et al., 2011*). Once cell cultures reached an $OD_{600}$ of 0.6, an untreated sample was harvested, and the remaining cells were split into four flasks. Twenty µg/ml cordycepin or an equivalent volume of DMSO (vehicle control) was added to each flask along with 500 µM 4tU or an equivalent volume of DMSO (unlabelled vehicle control). Samples were taken after 20 and 40 min. Due to the light-sensitive nature of 4tU, samples were treated and total RNA extracted in a light-proof setting.

Two µg total RNA was combined with 0.84 µl 3 M pH 5.2 sodium acetate, 0.2 µl 0.5 M EDTA, 1.3 µl TFEA and 1.3 µl 192 mM $NaIO_4$ and made up to 25 µl with RNAse-free $dH_2O$. Samples were incubated at 45°C for 1 hr prior to cooling on ice. $NaIO_4$ was precipitated with 300 mM sodium acetate pH 5.2 and 300 mM potassium chloride for 10 min on ice then centrifuged at top speed for 30 min at 4°C in a bench-top centrifuge. The supernatant was transferred to a fresh tube and RNA was precipitated with 2.5 volumes of 100% ethanol and 0.5 µl 5 mg/mL glycogen for 1 hr on ice. These were centrifuged at top speed for 30 min at 4°C. Supernatant was removed and pellets were washed with 80% ethanol and centrifuged at top speed for a further 10 min. Pellets were air-dried at room temperature and resuspended in 18 µl RNase-free $dH_2O$.

RNA samples were combined with 2 µl 10x reducing mix (0.2 µl 1 M Tris-HCl pH 7.5, 0.2 µl 1 M DTT, 0.4 µl 5 M NaCl, 0.04 µl 0.5 M EDTA pH 8.0) and incubated at 37°C for 30 min. These were then cleaned using Zymo RNA Clean and Concentrator-5 kit per manufacturer's instructions for total RNA and eluted in 15 µl of $dH_2O$.

An altered version of the mPAT assay involving a mTVN-PAT reverse primer rather than the mPAT reverse was used. This primer has an identical primer sequence to the mPAT reverse except for the addition of two 3' variable bases, V (A, G or C) and N (any base). These variable bases lock the primer to the polyadenylation site for reverse transcription.

To create mTVN-PAT cDNA, 400 ng of total RNA and 1 µl of the mTVN-PAT reverse were combined and brought to a volume of 12 µl with $dH_2O$ in 200 µl PCR tubes. Mixtures were incubated at 80°C for 5 min then cooled to 42°C for 1 min. Once cooled, tubes were flash-centrifuged and 7 µl of master mix was added containing 4 µl 5x SuperScript III buffer, 1 µl 100 mM DTT, 1 µl 10 mM dNTPs, and 1 µl RNaseOut per reaction. Samples were mixed thoroughly by inversion then flash-centrifuged and held at 42°C for 1 min. At this temperature, 1 µl of SuperScript III was added followed by mixing by inversion and flash-centrifuging. Samples were then further incubated at 42°C for 15 min, 48°C for 15 min and 55°C for 15 min followed by inactivation of reverse transcription at 80°C for 5 min and cooling to 12°C.

PCR amplification, sequencing and data processing were carried out as for the mPAT with 21 and 12 gene-specific forward primers for the BY4741 and *rpb1-1* experiments respectively and primer sequences are indicated in *Table 2*.

For each read aligned to the target genes, every mismatch along this sequence compared to the reference was found and a list of SNPs created. From this, the total number of T to C nucleotide conversions, indicative of 4tU labelling, was calculated. The total reads (in reads per million) and those with T to C conversions for each read length was compared for each sample condition. Reads aligned to the *OM14* gene were plotted with values representative of two replicates.

**Table 2.** Oligonucleotide Primers.

| Primer | Sequence (5' – 3') |
|---|---|
| PAT-seq biotin end-extend | Biotin-CAGACGTGTGCTCTTCCGATCTTTTTTTTTTTTTTTTTTTT |
| PAT-seq splint A (200 µM) | CCCTACACGACGCTCTTCCG(rA)(rT)(rC)(rT) |
| PAT-seq Splint B (200 µM) | NNNNAGATCGGAAGAGCGTCGTGTAGGG |
| Illumina universal Rd1 forward (50 µM for mPAT) | AATGATACGGCGACCACCGAGATCTACACTCTTTCCCTACACGACGCTCTTCCG |
| PAT anchor reverse | GCGAGCTCCGCGGCCGCGTTTTTTTTTTTT |
| TVN-PAT anchor reverse | GCGAGCTCCGCGGCCGCGTTTTTTTTTTTTTVN |
| OM14 PAT forward | GGGTCTTTTGACGCTGGAC |
| mPAT reverse | CAGACGTGTGCTCTTCCGATCTTTTTTTTTTTTT |
| mPAT-TVN reverse | CAGACGTGTGCTCTTCCGATCTTTTTTTTTTTTTVN |
| AAD16 mPAT | CCTACACGACGCTCTTCCGATCTTGCGTTTGTGTAAGAAATATGC |
| ADE2 mPAT | CCTACACGACGCTCTTCCGATCTAGAAACTGTCGGTTACGAAGC |
| APQ12 mPAT | CCTACACGACGCTCTTCCGATCTGAAACGCCTCTGCTTACTCGG |
| ARG8 mPAT | CCTACACGACGCTCTTCCGATCTGGCTATTGAAGCGGTTTACG |
| ARP5 mPAT | CCTACACGACGCTCTTCCGATCTGAGACAGCAAACTGAAACGC |
| CHD1 mPAT | CCTACACGACGCTCTTCCGATCTGCTGATGGCAATGTACGAC |
| COX17 mPAT | CCTACACGACGCTCTTCCGATCTCTGACAGTCTGCCGACAACCA |
| CRN1 mPAT | CCTACACGACGCTCTTCCGATCTCGGCGGCGATAATAATGC |
| CST6 mPAT | CCTACACGACGCTCTTCCGATCTCTCGAGCTGCATCCTTTCTT |
| DBF2 mPAT | CCTACACGACGCTCTTCCGATCTTCAACTAGCACCTATGAACGC |
| ECM16 mPAT | CCTACACGACGCTCTTCCGATCTGCTTCCAGACCATCACAGG |
| ECM25 mPAT | CCTACACGACGCTCTTCCGATCTGCATATACGACAACAAAATACCC |
| END3 mPAT | CCTACACGACGCTCTTCCGATCTGCAGAAATCAATTGACACCGA |
| ENT1 mPAT | CCTACACGACGCTCTTCCGATCTGTGATTCTGTCATTCCAGTCCG |
| ERG8 mPAT | CCTACACGACGCTCTTCCGATCTGGTAGATAATAGTGGTCCATGTGA |
| GFD1 mPAT | CCTACACGACGCTCTTCCGATCTCACATGGACACTTTTAAGCACG |
| HSP26 mPAT | CCTACACGACGCTCTTCCGATCTGGTTTCTTCTCAAGAATCGTG |
| IMP2 mPAT | CCTACACGACGCTCTTCCGATCTGAGCCATTTTAGAATGAAAATCAGC |
| LOS1 mPAT | CCTACACGACGCTCTTCCGATCTGCAAGGTCAATAGCTTTCAGG |
| MRPL19 mPAT | CCTACACGACGCTCTTCCGATCTCGAGAAATTTTCAACAGACCTTCC |
| MRPL22 mPAT | CCTACACGACGCTCTTCCGATCTGCTGAGAAAGATGAACTGCTACTC |
| MSA1 mPAT | CCTACACGACGCTCTTCCGATCTGCATGTGAATGGAGTTGACCTTC |
| NOP16 mPAT | CCTACACGACGCTCTTCCGATCTGGCAACTACCAATTGATTACCA |
| NOT3 mPAT | CCTACACGACGCTCTTCCGATCTCAGTGCTAATGGCAGTATAATTTG |
| NUP159 mPAT | CCTACACGACGCTCTTCCGATCTGCTATATGTACGTTGTTAGTGCCG |
| OM14 mPAT | CCTACACGACGCTCTTCCGATCTGGTCTTTTGACGCTGGACGG |
| OM45 mPAT | CCTACACGACGCTCTTCCGATCTCTGGAGCTCGAAAAAGGAC |
| PDE2 mPAT | CCTACACGACGCTCTTCCGATCTTTCCTTTTGTGAAGTATTTGTGC |
| PNT1 mPAT | CCTACACGACGCTCTTCCGATCTCATGACATTATCTATGCTGTACATATTG |
| PRY2 mPAT | CCTACACGACGCTCTTCCGATCTGGATTCTTCTTTTCTAGGGTACGC |
| RCL1 mPAT | CCTACACGACGCTCTTCCGATCTGGTGTAACTTCACGGACAACT |
| RER2 mPAT | CCTACACGACGCTCTTCCGATCTGAGCAAGATAAATGAGTTCGC |
| RHO1 mPAT | CCTACACGACGCTCTTCCGATCTCAATCCCATTCCTTTTCTCA |
| RPF1 mPAT | CCTACACGACGCTCTTCCGATCTCCGTAAGAACCGTGGTCG |
| RSC4 mPAT | CCTACACGACGCTCTTCCGATCTGTCTTCCCATCATATGCATGT |

*Table 2 continued on next page*

*Table 2 continued*

| Primer | Sequence (5′ – 3′) |
| --- | --- |
| SLF1 mPAT | CCTACACGACGCTCTTCCGATCTGGTGAAATTAGCAGGCAGTTTG |
| SNF2 mPAT | CCTACACGACGCTCTTCCGATCTGCATGACAGAAGCGAGTGTATAG |
| SRP68 mPAT | CCTACACGACGCTCTTCCGATCTGGTTTCTTGGGCCTATTTGG |
| TIM54 mPAT | CCTACACGACGCTCTTCCGATCTCCAAGGAAGAGCCAGAATCAG |
| TOM70 mPAT | CCTACACGACGCTCTTCCGATCTGCAGCAATGACATTGACATCTCAC |
| TRP2 mPAT | CCTACACGACGCTCTTCCGATCTAATGATGTATAGCAGGATCCTGA |
| UBC9 mPAT | CCTACACGACGCTCTTCCGATCTGAATCCATCTTTCCCATTCTTCC |
| VTS1 mPAT | CCTACACGACGCTCTTCCGATCTGCATATACCAACACAGGGAACA |
| YET3 mPAT | CCTACACGACGCTCTTCCGATCTGTCGATGTGCAAAAGCCTACA |
| YRA1 mPAT | CCTACACGACGCTCTTCCGATCTACCGCCACTAGGTGACGC |
| YSC84 mPAT | CCTACACGACGCTCTTCCGATCTGGATGGGTTCCTTATTCAGC |
| MID2 mPAT | CCTACACGACGCTCTTCCGATCTCAAGGTAACGAATTATCACCACG |

All primers have a concentration of 100 µM unless stated.

## Polysome profiling

Prior to cell harvesting, 100 µg/ml cycloheximide was added and incubation continued for another minute to stall ribosomes on the mRNA. After removal of supernatant, pellets were resuspended in 5 ml breaking buffer (20 mM Tris-HCl pH 7.5, 100 mM KCl, 0.5% NP40, 2 mM MgCl$_2$, 200 µg/ml heparin, 100 µg/ml cycloheximide, 2 mM DTT, 0.5 mM PMSF) on ice and transferred to corex tubes. Pellets were then resuspended in 1 ml breaking buffer and 4 µl RNaseOut and approximately 500 µl worth of glass beads added. Cells were lysed by vortexing corex tubes 10 times for 1 min with 1 min on ice in between. Supernatant was then transferred to 2 ml tubes and stored at −80°C until ready for further use.

Linear sucrose gradients were prepared in Beckman Coulter 14 × 96 mm centrifuge tubes with sequential loading of sucrose stocks in gradient buffer (50 mM Tris-HCl pH 7.0, 50 mM NH$_4$Cl, 0.5% NP40, 4 mM MgCl$_2$, 1 mM DTT and 100 µg/ml cycloheximide). From top to bottom, tubes contained 2.5 ml 17.5%, 2.5 ml 25%, 2 ml 33%, 2 ml 41% and 2.5 ml 50% sucrose solutions. Lysate was loaded to the top of the sucrose gradients and spun in a swinging bucket rotor Beckman Coulter SW40TI in a L-90K Preparative Ultracentrifuge at 38000 rpm for 2.5 hr at 4°C, max acceleration, no break. Gradients were fractionated by upward displacement with 60% sucrose at 3 ml/min and absorbance was monitored at 254 nm.

## Proteomics

Cell pellets were resuspended in 25 µl 1% SDC (sodium deoxycholate), 100 mM Tris and broken open with cryo-pulverisation using a cryogenic mixer mill (50 small ball bearings per sample, 4 × 45 s at a frequency of 30 Hz with intermittent cooling in liquid nitrogen for 1 min). The protein content was estimated using the bicinchoninic acid method. Samples were denatured and alkylated for 5 min at 95°C with TCEP (Tris(2-carboxyethyl phosphine hydrochloride)) and CAA (2-chloroacetamide) at final concentrations of 10 mM TCEP and 40 mM CAA. Sequencing-grade trypsin was added at an enzyme:protein ratio of 1:100 and incubated overnight at 37°C. Protein digestion was arrested through the addition of formic acid to a final concentration of 1% and two-phase extraction with water-saturated ethyl acetate was used to remove SDC. The aqueous phase containing the peptides was collected, concentrated in a vacuum concentrator and reconstituted in buffer A (0.1% formic acid).

Data-independent acquisition (DIA) mass spectrometry was performed on an Orbitrap Fusion Tribrid mass spectrometer (Thermo Scientific) coupled to a Dionex UltiMate 3000 RSLCnano system and a Dionex UltiMate 3000 RS autosampler. The samples were loaded onto an Acclaim PepMap 100 trap column (100 µm x 2 cm; nanoViper; C18; 5 µm; 100 Å) and separated on an Acclaim PepMap RSLC analytical column (75 µm x 5 cm; nanoViper; C18; 2 µm; 100 Å). The parameters for the

mass spectrometric DIA acquisition were described previously (*Deo et al., 2020*). The acquired DIA data was analysed in Spectronaut 8 Cassini (Biognosys) using an in-house generated spectral library established by acquiring the same samples with the same set-up in data-dependent acquisition mode.

## TVN-PAT

TVN-PAT was performed as previously described (*Jänicke et al., 2012*). To create TVN-PAT cDNA, 1 µg of total RNA and 1 µl TVN primer were combined and brought to a volume of 12 µl with dH2O in 200 µl PCR tubes. Mixtures were incubated at 80℃ for 5 min then cooled to 42℃ for 1 min. Once cooled, tubes were flash-centrifuged and 7 µl of master mix was added containing 4 µl 5x Super-Script III buffer, 1 µl 100 mM DTT, 1 µl 10 mM dNTPs and 1 µl RNaseOut per reaction. Samples were mixed thoroughly by inversion then flash-centrifuged and held at 42℃ for 1 min. At this temperature, 1 µl of SuperScript III was added followed by mixing by inversion and flash-centrifuging. Samples were then further incubated at 42℃ for 15 min, 48℃ for 15 min and 55℃ for 15 min followed by inactivation of reverse transcription at 80℃ for 5 min and cooling to 12℃.

For PCR amplification, cDNA was diluted 1:10 with the addition of 200 µl of $dH_2O$. 5 µl diluted cDNA was added to fresh 200 µl PCR tubes with 15 µl master mix containing 0.2 µl PAT assay reverse primer, 0.2 µl *OM14* gene-specific forward primer, 10 µl Amplitaq Gold 360 Master Mix and 4.6 µl $dH_2O$ per reaction. Amplification was performed with cycle conditions of 95℃ for 10 min, then 28 cycles of 95℃ for 20 s, 60℃ for 20 s and 72℃ for 30 s, followed by 1 min at 72℃.

Four µl 6x Orange G loading dye was added to each PCR sample. 12 µl was run on a 2% high resolution agarose gel pre-stained with SYBR safe. To estimate PCR product sizes, band migration was determined relative to a 100 bp ladder.

## Seahorse metabolic analysis

BY4741 cells were subjected to Seahorse XF Cell Energy Phenotype assay per the manufacturer's instructions (Agilent) with modifications as follows. Cells were grown at 30℃ to $OD_{600}$ 0.6 in synthetic media then resuspended in assay medium (Seahorse XF media, 20 µg/ml uracil, 5 mg/ml glucose, 5 µg/ml ammonium sulphate) and plated (5 × 104 per well) in CellTak-coated XF96 plates via centrifugation (1000 g, 5 min, RT). After 1 hr recovery at 30℃, Oxygen Consumption Rate (OCR) and Extracellular Acidification Rate (ECAR) were determined using an XF96 extracellular flux analyser (Seahorse Biosciences) at 30℃. Using cycling parameters 1 min mix, 1 min wait, 2 min measure, basal activity was measured for four cycles. After addition of 20 µg/ml cordycepin, or an equivalent volume of DMSO or assay media (controls), a further 10 cycles were measured before uncoupling of mitochondrial function via stressor cocktail (3 µM FCCP, 1 µM oligomycin), followed by a further six measurement cycles. Data are representative of three independent experimental repeats, with 12 wells per condition in each. Graphs were prepared using Prism seven software.

## Acknowledgements

We thank the following researchers for strain and plasmid resources: David Brow, Michael Culbertson, Walter Keller, Lionel Minvielle-Sebastia, Beate Schwer, Mike Stark, Maurice Swanson, Françoise Wyers, Craig Kaplan and Wendy Olivas. Melissa J Curtis is acknowledged for her technical assistance in the experimental data collection early in the project, and members of the Beilharz laboratory provided critical feedback. The MHTP Medical Genomics Facility, the Monash Proteomics and Metabolomics Facility, Micromon, and the Monash Bioinformatics Platform are thanked for the provision of technical support and infrastructure. Monash research technology platforms are enabled by Bioplatforms Australia (BPA) and the National Collaborative Research Infrastructure Strategy (NCRIS). RET was supported by an Australian Postgraduate Research award. THB was supported by a Monash Bio Discovery Fellowship and grants from the Australian Research Council (ARC: DP170100569 and FT180100049).

## Additional information

### Funding

| Funder | Grant reference number | Author |
|---|---|---|
| Australian Research Council | DP170100569 | Traude H Beilharz |
| Australian Research Council | FT180100049 | Traude H Beilharz |

The funders had no role in study design, data collection and interpretation, or the decision to submit the work for publication.

### Author contributions
Rachael Emily Turner, Data curation, Formal analysis, Validation, Visualization, Methodology, Writing - original draft; Paul F Harrison, Data curation, Formal analysis, Supervision, Visualization, Writing - review and editing; Angavai Swaminathan, Supervision, Investigation, Methodology, Writing - review and editing; Calvin A Kraupner-Taylor, Data curation, Formal analysis, Investigation, Methodology, Writing - review and editing; Belinda J Goldie, Amanda L Peterson, Data curation, Formal analysis, Methodology, Writing - review and editing; Michael See, Data curation, Formal analysis, Supervision, Writing - review and editing; Ralf B Schittenhelm, Resources, Data curation, Formal analysis, Visualization, Methodology, Writing - review and editing; David R Powell, Resources, Data curation, Supervision, Writing - review and editing; Darren J Creek, Resources, Data curation, Formal analysis, Methodology, Writing - review and editing; Bernhard Dichtl, Conceptualization, Resources, Formal analysis, Writing - original draft, Writing - review and editing; Traude H Beilharz, Conceptualization, Resources, Supervision, Funding acquisition, Investigation, Project administration, Writing - review and editing

### Author ORCIDs
Rachael Emily Turner ⓘ https://orcid.org/0000-0001-9319-4825
Paul F Harrison ⓘ http://orcid.org/0000-0002-3980-268X
Michael See ⓘ http://orcid.org/0000-0002-7231-3896
Traude H Beilharz ⓘ https://orcid.org/0000-0002-8942-9502

### Decision letter and Author response
Decision letter https://doi.org/10.7554/eLife.65331.sa1
Author response https://doi.org/10.7554/eLife.65331.sa2

## Additional files

### Supplementary files
• Transparent reporting form

### Data availability
Sequencing data have been deposited in GEO under accession code GSE160539.

The following dataset was generated:

| Author(s) | Year | Dataset title | Dataset URL | Database and Identifier |
|---|---|---|---|---|
| Turner RE, Harrison PF, Swaminathan A, Kraupner-Taylor CA, Goldie BJ, See M, Peterson AL, Schittenhelm RB, Powell DR, Creek DJ, Dichtl B, Beilharz TH | 2021 | Genetic and pharmacological evidence for kinetic competition between alternative poly(A) sites in yeast | https://www.ncbi.nlm.nih.gov/geo/query/acc.cgi?&acc=GSE160539 | NCBI Gene Expression Omnibus, GSE160539 |

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
