## [Decision Letter]

**Acceptance summary:**

This study aims to provide a comprehensive analysis of factors governing polyadenylation site selection in yeast. Overall, the authors reveal that multiple but distinct inputs including polyadenylation machinery integrity, transcription elongation rate, nucleotide availability and chromatin landscape all contribute to controlling cleavage and polyadenylation. The work provides the field with an understanding of how aspects of cellular state can control mRNA 3' end formation.

**Decision letter after peer review:**

Thank you for submitting your article "Genetic and pharmacological evidence for kinetic competition between alternative poly(A) sites in yeast" for consideration by *eLife*. Your article has been reviewed by 3 peer reviewers, one of whom is a member of our Board of Reviewing Editors, and the evaluation has been overseen by James Manley as the Senior Editor. The reviewers have opted to remain anonymous.

Essential Revisions:

Overall, the sentiments of the individual Reviewers were quite overlapping. In a broad sense, the most important task to address through revisions is to improve the connections between the individual elements of the study. Reviewers judged there were some reasonable experiments that could be done to mitigate the disparateness this fact, in addition to adjustments to the text. Among the suggested revisions, the following stood out in discussion, but the authors should also consider the reviews in aggregate as there were a substantial number of concerns:

1. The authors conclude that transcription elongation impacts APA based on MPA treatment and the slow RNAPII mutant rpb1-H1085Y promoting proximal poly(A) site usage in cells treated with cordycepin. To build support for this model, the authors should examine whether MPA treatment and/or the rpb1 mutant cause proximal poly(A) site usage and rescue in some of the 3'-end processing mutants examined.

2. Can the authors utilize temperature sensitive mutants in some of the nucleotide biosynthesis enzymes to either mask the cordecypin effect or, if there are gain-in-function mutations can they show the PAS shift in the absence of cordecypin?

3. Can the authors test some of the defined RNAPII mutants that have been found to be increased in elongation efficiency? Do these cause a shift on their own?

*Reviewer #1 (Recommendations for the authors):*

1. In figure one, the authors describe the PAT-seq data as representative of the mutant collection. But from close inspection, it appears that the one non-temperature sensitive strain (Clp1-pm) is the only strain that didn't exhibit a significant PAS shift but all of the temp sensitive strains did. Further, the authors do not show what happened to the other two strains that did not exhibit temperature sensitivity (yth1 and syc1). This leads to an important question: is the PAS shift simply due to yeast being shifted to a restrictive temperature that they cannot tolerate? Can the authors address this?

2. Conceptually, there is an abrupt pivot between figures 3 and 4. The authors uncover an interesting phenotype with Sen1 mutant but then explore the role of cordecypin in PAS choice. I am having trouble understanding the connection. In my mind, the next figure should have investigated the mechanism of sen1 termination to understand the opposing phenotypes

3. The authors arrive at a conclusion that cordecypin treatment triggers upregulation of nucleotide biosynthesis, which leads to enhances transcription elongation rate. This seems to be an important conclusion as it opens up the final segment of the paper and is critical to their model. Can the authors utilize temperature sensitive mutants in some of the nucleotide biosynthesis enzymes to either mask the cordecypin effect or, if there are gain-in-function mutations can they show the PAS shift in the absence of cordecypin?

4. The experiment of showing a slow RNAPII mutant to dampen the cordecypin treatment is a very nice result. Can the authors test some of the defined RNAPII mutants that have been found to be increased in elongation efficiency? Do these cause a shift on their own?

*Reviewer #2 (Recommendations for the authors):*

Other comments to improve the quality of the paper:

The Introduction lacks information on cordycepin.

It is not clear if the authors found a role for NAB4 in APA (compare Figure 1 and Figure 2).

Validation of a few genes by conventional methods other than Seq could be done.

Line 108 – Sup Figure 1 in the text also refers to cordycepin, which is missing from text.

Line 134 related to Figure 1 C – rna14 and pcf11 mutants cause a complete switch in SUB2 APA, but there is no complete switch for the other mutants used and there is still growth for those mutants (Figure 1B). However, by PAT-Seq all all mutants with the exception of clp1-pm cause defects in APA, leading to longer 3'UTRs. How do you reconcile these observations? Clp1-pm does not have an effect – this should be highlighted. What is its function in pA?

It is interesting that the mutant where the authors show a bigger effect in APA is for ysh1. The authors suggest that proximal PAS is skipped because the cleavage reaction is inefficient. Does this APA switch depends on PAS strength?

167 – how are these APA numbers different from previous published data?

Sup Figure 2 refers to cordycepin, not referred to in the text.

line 187 – nab4 does not show an effect as stronger as rna14 or pcf11, therefore the suggested role for CFIB in the cleavage reaction should be toned down. It is not known if the PAS switch is due to a defect in cleavage or PAS recognition.

line 91 – cleavage recognition or PAS recognition?

line 192 – "usage of longer" should be substituted by "higher expression of longer"

line 208 – syc does not look like the only non-essential gene in Figure 1B

Figure 2 B and line 225-231 – this part is repetitive for yth1. The function of nab4 in OM45 APA is not highlighted. On the other hand, how do the authors reconcile the nab4 result with the lack of an effect shown in Figure 1B?

Figure 2 B is confusing with the red rectangle positioned at zero for yth1 graph only.

line 253 and Figure 3 C – For all genes except UBX6 sen1-1 causes a reduction in reads but it is referred that there is no difference in transcript levels (data not shown). Data need to be shown. GAC1 is not representative of 3'UTR shortening.

*Reviewer #3 (Recommendations for the authors):*

1. The authors conclude that six of the 3'-end processing mutants (rna14-1, pcf11-2, nab4-1, ysh1-13, fip1-1, pap1-1) cause a switch to more distal cleavage site usage and longer 3'-UTRs, but little change is observed in clp1-pm, based on the PAT-seq data. They suggest that the clp1-pm mutant, containing point mutations that affect ATP binding, does not impact poly(A) site selection. Given the similarity in the global 3'-end shift effects in Figure 1D, one could conclude that inactivation of the six 3'-end processing factors, regardless of subcomplex, impacts poly(A) site usage in a similar manner, predominantly increasing distal site selection. However, an alternative explanation is that all six 3'-end processing factor mutants are similarly unstable at 37{degree sign}C and similarly decrease the stability/integrity of the core cleavage and polyadenylation complex resulting a similar APA profile. In the case of clp1-pm, this mutant might not alter APA because it does not impact the stability/integrity of the core complex. The authors should therefore examine the levels/stability of cleavage and polyadenylation complex proteins in these 3'-end processing mutants. Notably, mass spectrometry was performed on cells treated with cordycepin to address this possibility for the drug. In addition, the authors could test other alleles of 3'-end processing mutants that are documented not to significantly alter the levels of the core cleavage and polyadenylation complex. These experiments may expose differences between the 3'-end processing factors in poly(A) site selection.

2. The sen1-1 mutant is shown to cause significant switching to shorter 3'-UTRs for 676 genes, suggesting Sen1 antagonizes cleavage site choice by the core 3'-end processing machinery at a subset of protein-coding genes. Given that 411 genes were altered in cleavage site usage in both sen1-1 and pcf11-2 mutants in opposing directions, it would informative to assess whether a sen1-1 pcf11-2 double mutant exhibits a positive genetic interaction and restores growth of the cells and some of the altered poly(A) site usage back to wildtype.

3. The 3'-end processing factor mutants are all generated in the W303 strain background, whereas cells treated with cordycepin are all in the BY4741 strain background. Since several comparisons of the PAT-seq data on the 3'-end processing mutants are made with PAT-seq data on the cells treated with cordycepin, the authors should perform some experiments on W303 cells treated with cordycepin to validate that poly(A) site changes observed in the BY4741 strain are indeed similarly seen in the W303 strain.

4. The authors propose a kinetic model for poly(A) site selection involving a balance between the concentration of 3'-end processing factors and transcription elongation. The authors conclude that transcription elongation impacts APA based on MPA treatment and the slow RNAPII mutant rpb1-H1085Y promoting proximal poly(A) site usage in cells treated with cordycepin. To build support for this model, the authors should examine whether MPA treatment and/or the rpb1 mutant cause proximal poly(A) site usage and rescue in some of the 3'-end processing mutants examined.

5. The nab3-11 growth at 25{degree sign}C in Figure 1B is extremely impaired, suggesting this temperature-sensitive mutant is very sick at 25{degree sign}C. This observation is inconsistent with the published literature, where nab3-11 growth is similar to wildtype at 25{degree sign}C. The authors should repeat the nab3-11 spotting.

6. The Y-axis scales/data ranges for wildtype and 3'-end processing mutants in Figure 1C, Figure 3C, and Figure 4A are all different in order to display the changes in poly(A) site selection/length of 3'-UTR. If the sequence reads are normalized, would it not be possible to select a convenient, identical Y-axis scale/data range for the wildtype and mutants, so the depth of read coverage is better represented.

7. The authors compare their PAT-seq dataset on cells treated with cordycepin to a publicly available dataset of yeast nucleosome occupancy and draw the conclusion that cordycepin mediated usage of distal poly(A) sites is associated with a permissive chromatin template containing less nucleosomes. Could the authors comment on the similarities and differences between BY4741 cells treated with cordycepin used in this study and the cells used for nucleosome occupancy. Differences in strain or growth conditions between the datasets could impact the comparison and conclusions. It would be interesting but not immediately essential to test whether cordycepin responsive genes change their APA profile in a nucleosome mutant.

---

## [Author Response]

Essential Revisions:Overall, the sentiments of the individual Reviewers were quite overlapping. In a broad sense, the most important task to address through revisions is to improve the connections between the individual elements of the study. Reviewers judged there were some reasonable experiments that could be done to mitigate the disparateness this fact, in addition to adjustments to the text. Among the suggested revisions, the following stood out in discussion, but the authors should also consider the reviews in aggregate as there were a substantial number of concerns:

The essential revisions have been addressed such that we now show two additional lines of evidence to consolidate the connection between transcriptional rate and APA (Figure 8). New figures have been included that show APA depends on nucleotide availability and that RNA Pol II fast alleles can indeed drive 3’UTR lengthening. We have made further changes throughout the manuscript to incorporate reviewer comments as applicable. Some misunderstandings have been clarified here without further change to the manuscript.

1. The authors conclude that transcription elongation impacts APA based on MPA treatment and the slow RNAPII mutant rpb1-H1085Y promoting proximal poly(A) site usage in cells treated with cordycepin. To build support for this model, the authors should examine whether MPA treatment and/or the rpb1 mutant cause proximal poly(A) site usage and rescue in some of the 3'-end processing mutants examined.

To further probe the connection between genetic and pharmacological sources of APA, we treated *pcf11-2* and *sen1-1* cells with cordycepin and MPA. These data were included in a new panel, figure 6G. Cordycepin treatment has little effect on *pcf11-2* and *sen1-1* cells. Once 3’ UTRs are lengthened in *pcf11-2*, an increase in transcriptional rate cannot further influence 3’ end choice. If Cordycepin treatment can override the *sen1-1*, its effect is subtle. However, MPA was able to suppress the *pcf11-2* mutant phenotype and exacerbate shortening of 3’UTRs in *sen1-1* mutants. This indicates that a re-balancing of transcription to available cleavage and polyadenylation machinery (see figure 8) can normalise 3’ end choice. The text has been updated to include these data.

2. Can the authors utilize temperature sensitive mutants in some of the nucleotide biosynthesis enzymes to either mask the cordecypin effect or, if there are gain-in-function mutations can they show the PAS shift in the absence of cordecypin?

The pharmacological version of this was shown with MPA, which is an inhibitor of inosine monophosphate dehydrogenase (IMPDH). MPA treatment, causing a reduction in GTP, was able to suppress cordycepin’s effect on APA. No suitable temperature-sensitive mutant was identified to look at this genetically. Moreover, given the high level of enzymatic redundancy and feedback control that maintains cellular nucleotide homeostasis, we do not think that in vivo modulation of nucleotide levels can be robustly achieved in a reasonable timeframe.

However, to address the question and provide further support for the finding with MPA (that limiting nucleotides limit 3’ UTR extension), we exposed exponentially growing W303 cells, which are adenine auxotrophs, to media with or without supplemented adenine. We found that the absence of adenine promoted 3’ UTR shortening, without an overt decrease in mRNA abundance in the timeframe tested. Thus, adenine withdrawal affected APA in the opposite direction compared to cordycepin. This finding was included in a new data panel figure 6E. This new data provides a second layer of evidence to connect limiting nucleotides to 3’ UTR shortening. The text has been updated accordingly.

3. Can the authors test some of the defined RNAPII mutants that have been found to be increased in elongation efficiency? Do these cause a shift on their own?

This was effect was recently shown by Geisberg et al. (2020). We repeated the experiment using two different fast RNA polymerase II alleles and confirmed a switch to a more distal cleavage site for the gene *OPI3* when compared to the wild-type as previously shown (Geisberg et al. 2020, Figure 4A). This is now included as a new panel, Figure 6C and is reflected by updated text.

We are confident our manuscript provides robust evidence for the nexus between the level of functional cleavage and polyadenylation machinery, nucleotide concentration and transcriptional rate. Figure 8 has been updated to reflect the additional data that was generated by addressing these essential revisions.

Reviewer #1 (Recommendations for the authors):1. In figure one, the authors describe the PAT-seq data as representative of the mutant collection. But from close inspection, it appears that the one non-temperature sensitive strain (Clp1-pm) is the only strain that didn't exhibit a significant PAS shift but all of the temp sensitive strains did. Further, the authors do not show what happened to the other two strains that did not exhibit temperature sensitivity (yth1 and syc1). This leads to an important question: is the PAS shift simply due to yeast being shifted to a restrictive temperature that they cannot tolerate? Can the authors address this?

Our APA results have been normalised to wild-type (see 135 and 182). As such the observed effects are not the result of the temperature shift. Also, note that *yth1-1* and *yth1-4* are formamide *ts* strains (Barabino et al. 1997; Barabino et al. 2000). While the strains do not display *ts* lethal growth in the absence of formamide the biochemical phenotypes are well established at 37^o^C in the absence of formamide (Figure 2A). The *Dsyc1* phenotype, ie no effect on APA, is shown in Figure 2A.

2. Conceptually, there is an abrupt pivot between figures 3 and 4. The authors uncover an interesting phenotype with Sen1 mutant but then explore the role of cordecypin in PAS choice. I am having trouble understanding the connection. In my mind, the next figure should have investigated the mechanism of sen1 termination to understand the opposing phenotypes

We have added a connecting sentence to improve the flow between figures 3 and 4 (226-229). “In sum, these genetic data indicate that APA hinges on the level of functional machineries for cleavage and polyadenylation. However, given that each temperature sensitive allele has a unique sensitivity and severity and therefore a different degree of apparent impact on 3’ end choice, we next sought to explore the impact of cordycepin as a possible pharmacological driver of APA.”

We agree, the NNS data open interesting new avenues of research connecting Sen1 to termination. However, given that the manuscript already represents a large body of work, and that first author Rachael Turner’s PhD candidature must come to an end, any further investigation is beyond the scope of the current manuscript.

3. The authors arrive at a conclusion that cordecypin treatment triggers upregulation of nucleotide biosynthesis, which leads to enhances transcription elongation rate. This seems to be an important conclusion as it opens up the final segment of the paper and is critical to their model. Can the authors utilize temperature sensitive mutants in some of the nucleotide biosynthesis enzymes to either mask the cordecypin effect or, if there are gain-in-function mutations can they show the PAS shift in the absence of cordecypin?

See our response to ‘essential revisions’ comment 2 above.

4. The experiment of showing a slow RNAPII mutant to dampen the cordecypin treatment is a very nice result. Can the authors test some of the defined RNAPII mutants that have been found to be increased in elongation efficiency? Do these cause a shift on their own?

See our response to ‘essential revisions’ comment 3 above.

Reviewer #2 (Recommendations for the authors):Other comments to improve the quality of the paper:The Introduction lacks information on cordycepin.

We did not include cordycepin in the introduction reasoning that its inclusion would be confusing. Previous literature mainly connects cordycepin to RNA chain termination and the inhibition of cytoplasmic polyadenylation a topic that would only serve to complicate the manuscript.

It is not clear if the authors found a role for NAB4 in APA (compare Figure 1 and Figure 2).

*nab4-1* shows considerable APA in figure 1 with 1945 genes undergoing significant changes. Both *nab4-1* and *nab4-7* show APA in figure 2A with the majority of genes having effect size scores greater than absolute 0.1. For the gene *OM45*, *nab4-7* has a greater impact on APA than *nab4-1*. This likely indicates the severity of the mutation.

Validation of a few genes by conventional methods other than Seq could be done.

Figure 6 now includes several genes looked at using traditional PCR and gel electrophoresis. In our hands, the vast majority of APA identified by our bioinformatic pipeline can be easily validated by conventional methods. The confounding effects of changing expression levels on band intensity, however, means that it is no longer our preferred choice for data visualisation.

Line 108 – Sup Figure 1 in the text also refers to cordycepin, which is missing from text.

There seems to have been confusion here between Supplemental figure and file. The text is correct.

Line 134 related to Figure 1 C – rna14 and pcf11 mutants cause a complete switch in SUB2 APA, but there is no complete switch for the other mutants used and there is still growth for those mutants (Figure 1B). However, by PAT-Seq all all mutants with the exception of clp1-pm cause defects in APA, leading to longer 3'UTRs. How do you reconcile these observations? Clp1-pm does not have an effect – this should be highlighted. What is its function in pA?

The effect size graphs shown in figure 1D indicate the extent of switching. Genes in which most transcripts are now using a more distal cleavage site in the mutant will therefore have larger effect size scores than those that use a mix of both cleavage sites. As such, *rna14-1* and *pcf11-2* genes tend to have higher effect scores. Growth does not indicate whether a mutation is active, it only indicates whether it is severe enough to interfere with cell viability.

The mutant strains used have different temperature sensitivities and severities; however, as stated in the text, common growth conditions were used to aid in comparison of results. Alternative polyadenylation is significantly occurring in all the mutants, however, the extent of switching differs and, as shown by the higher effect size points in *pcf11-2* and *rna14-1* as well as the IGV plot for *SUB2*, there is a greater degree of switching in these two mutants. The text already states that “Little change was noticeable for the clp1-pm mutant” and its function is addressed by the text “suggesting that ATP binding by Clp1 was dispensable for poly(A) site selection. Consistently, the mutations have previously been shown to have limited impact on pre-mRNA 3’ end formation and termination in vitro (Holbein et al., 2011)”. To dissect the role of Clp1 function in APA would require the generation of new temperature sensitive allele(s), which is beyond the scope of the current study.

It is interesting that the mutant where the authors show a bigger effect in APA is for ysh1. The authors suggest that proximal PAS is skipped because the cleavage reaction is inefficient. Does this APA switch depends on PAS strength?

The cleavage reaction is likely inefficient because Ysh1 is thought to be the endonuclease responsible for transcript cleavage. The idea of PAS strength comes from systems where the PAS consensus sequence and its non-canonical counterparts are well defined. In yeast, the PAS is degenerate and has not, to our knowledge, been systematically analysed for strength. Our data would argue that transcriptional rate and chromatin context are the main drivers of cleavage site choice in yeast.

167 – how are these APA numbers different from previous published data?

The number of genes undergoing APA in our data (~70%) are very similar to other published studies that indicate ~70% or ~80% of genes having detectible APA at steady state (Ozsolak et al. 2010) or under nutrient stress (Geisberg et al. 2020) respectively. Any differences in number are due to differences in the bioinformatic algorithms used to detect discrete alternative forms and to quantify their differential expression.

Sup Figure 2 refers to cordycepin, not referred to in the text.

Data in Sup Figure 2, renamed Figure 4 —figure supplement 2, is referred to from lines 265-312 and mentions cordycepin. If Supplementary file 2, renamed Figure 4 – Source Data 1, was meant, this file’s referral to cordycepin is now included in the Figure 4 legend.

line 187 – nab4 does not show an effect as stronger as rna14 or pcf11, therefore the suggested role for CFIB in the cleavage reaction should be toned down. It is not known if the PAS switch is due to a defect in cleavage or PAS recognition.

Nab4 mutation causes significant APA in 1945 genes which is more than *pcf11-2.* It is important to understand that the extent of switching is not “all or nothing”. To be detected as a shift, there needs to be a change in expression ratio of the two major 3’ UTR isoforms. This can come from a change in expression, PAS recognition, cleavage, polyadenylation or transcript stability. Our data can only report on the absolute abundance ratio of adenylated 3’ ends.

line 91 – cleavage recognition or PAS recognition?

Both presumably. PAS recognition is less clear cut in yeast than in other systems as detailed above.

line 192 – "usage of longer" should be substituted by "higher expression of longer"

The wording has been changed (170).

line 208 – syc does not look like the only non-essential gene in Figure 1B

All genes analysed, with the sole exception of *SYC1*, are essential for cell viability when deleted in the haploid genome (see *Saccharomyces cerevisiae* Genome Database). Because of this we analyse predominantly conditional (temperature sensitive) mutants. The degree of temperature sensitivity depends on the extent of damage/loss of protein function.

Figure 2 B and line 225-231 – this part is repetitive for yth1. The function of nab4 in OM45 APA is not highlighted. On the other hand, how do the authors reconcile the nab4 result with the lack of an effect shown in Figure 1B?

Figure 2B highlights differences in cleavage site choice for different *yth1* and *nab4* alleles, respectively. Figure 1D (rather than 1B) clearly shows a shift to longer 3’ UTRs for *nab4-1*. The impact of Nab4 on OM45 is described line 191-195.

Figure 2 B is confusing with the red rectangle positioned at zero for yth1 graph only.

We feel that it would be repetitive to have a schematic under every graph and that doing so would make these too small to be useful. The figure legend has been updated to help with interpretation.

line 253 and Figure 3 C – For all genes except UBX6 sen1-1 causes a reduction in reads but it is referred that there is no difference in transcript levels (data not shown). Data need to be shown. GAC1 is not representative of 3'UTR shortening.

The height of the peak in these visualisations represents the relative abundance of transcript isoforms. The indicated transcripts in *sen1-1* have fewer reads aligned to the distal form relative to the proximal form when compared to the wild-type. This is 3’UTR shortening. For *GAC1* the ratio of short isoform relative to the long isoform is statistically significantly changed from the ratio in the wild-type (note changed peak ratio). Data are scaled to emphasise this ratio, absolute differences in transcript levels take into account the total number of reads that align to a specific locus (i.e. short isoform + long isoform).

Reviewer #3 (Recommendations for the authors):1. The authors conclude that six of the 3'-end processing mutants (rna14-1, pcf11-2, nab4-1, ysh1-13, fip1-1, pap1-1) cause a switch to more distal cleavage site usage and longer 3'-UTRs, but little change is observed in clp1-pm, based on the PAT-seq data. They suggest that the clp1-pm mutant, containing point mutations that affect ATP binding, does not impact poly(A) site selection. Given the similarity in the global 3'-end shift effects in Figure 1D, one could conclude that inactivation of the six 3'-end processing factors, regardless of subcomplex, impacts poly(A) site usage in a similar manner, predominantly increasing distal site selection. However, an alternative explanation is that all six 3'-end processing factor mutants are similarly unstable at 37{degree sign}C and similarly decrease the stability/integrity of the core cleavage and polyadenylation complex resulting a similar APA profile. In the case of clp1-pm, this mutant might not alter APA because it does not impact the stability/integrity of the core complex. The authors should therefore examine the levels/stability of cleavage and polyadenylation complex proteins in these 3'-end processing mutants. Notably, mass spectrometry was performed on cells treated with cordycepin to address this possibility for the drug. In addition, the authors could test other alleles of 3'-end processing mutants that are documented not to significantly alter the levels of the core cleavage and polyadenylation complex. These experiments may expose differences between the 3'-end processing factors in poly(A) site selection.

Our analysis reveals a switch to distal cleavage sites not only for six (Figure 1D), but for twenty mutants in the core machinery (Figure 2A). The phenotypes of most mutants have previously been characterised by the scientific community in great detail and are well documented in the literature (see Table 1). Importantly, the majority of mutant alleles display distinct and specific defects, eg, in pre-mRNA cleavage activity, pre-mRNA polyadenylation, 3’ end cleavage site recognition, RNA binding or transcription termination. As such it is not to be expected that the impact of the mutants analysed here is due to a common destabilisation of the core machinery. The text has been updated see response to reviewer 1.

2. The sen1-1 mutant is shown to cause significant switching to shorter 3'-UTRs for 676 genes, suggesting Sen1 antagonizes cleavage site choice by the core 3'-end processing machinery at a subset of protein-coding genes. Given that 411 genes were altered in cleavage site usage in both sen1-1 and pcf11-2 mutants in opposing directions, it would informative to assess whether a sen1-1 pcf11-2 double mutant exhibits a positive genetic interaction and restores growth of the cells and some of the altered poly(A) site usage back to wildtype.

The role of Sen1 in APA has not been further investigated here. The suggested genetic analysis of the *sen1-1 pcf11-2* interaction, or the interaction of *sen1-1* with any other of the analysed mutants that cause 3 UTR lengthening, should be considered in future analysis of the *sen1-1* phenotype. We feel, however, that this insight is not essential for the overall conclusions drawn in this work. To the contrary, given Sen1’s established role in non-coding RNA metabolism, we suggest it is unlikely to restore growth but could instead further enfeeble the *pcf11-2* strain, and further complicate the manuscript.

3. The 3'-end processing factor mutants are all generated in the W303 strain background, whereas cells treated with cordycepin are all in the BY4741 strain background. Since several comparisons of the PAT-seq data on the 3'-end processing mutants are made with PAT-seq data on the cells treated with cordycepin, the authors should perform some experiments on W303 cells treated with cordycepin to validate that poly(A) site changes observed in the BY4741 strain are indeed similarly seen in the W303 strain.

An example of cross-strain conservation of APA was carried out for the OM14 gene and is included in figure 6G.

4. The authors propose a kinetic model for poly(A) site selection involving a balance between the concentration of 3'-end processing factors and transcription elongation. The authors conclude that transcription elongation impacts APA based on MPA treatment and the slow RNAPII mutant rpb1-H1085Y promoting proximal poly(A) site usage in cells treated with cordycepin. To build support for this model, the authors should examine whether MPA treatment and/or the rpb1 mutant cause proximal poly(A) site usage and rescue in some of the 3'-end processing mutants examined.

See our response to ‘essential revisions’ comments 1, 2 and 3 above.

5. The nab3-11 growth at 25{degree sign}C in Figure 1B is extremely impaired, suggesting this temperature-sensitive mutant is very sick at 25{degree sign}C. This observation is inconsistent with the published literature, where nab3-11 growth is similar to wildtype at 25{degree sign}C. The authors should repeat the nab3-11 spotting.

We have repeated the *nab3-11* drops and revised Figure 1B accordingly.

6. The Y-axis scales/data ranges for wildtype and 3'-end processing mutants in Figure 1C, Figure 3C, and Figure 4A are all different in order to display the changes in poly(A) site selection/length of 3'-UTR. If the sequence reads are normalized, would it not be possible to select a convenient, identical Y-axis scale/data range for the wildtype and mutants, so the depth of read coverage is better represented.

We agree that this is a conundrum. However, when the Y-axis is fixed as suggested, the visualisation is still confounded because, where read-coverage is distributed over two peaks, total expression appears reduced compared to a single peak. As part of the bioinformatic processing, global expression is normalised for statistical testing. In our view, there is something to be said for showing raw data too. And since the expression ratio of 3’ end isoforms is the question at hand, we settled on a visualisation that best illustrated that.

7. The authors compare their PAT-seq dataset on cells treated with cordycepin to a publicly available dataset of yeast nucleosome occupancy and draw the conclusion that cordycepin mediated usage of distal poly(A) sites is associated with a permissive chromatin template containing less nucleosomes. Could the authors comment on the similarities and differences between BY4741 cells treated with cordycepin used in this study and the cells used for nucleosome occupancy. Differences in strain or growth conditions between the datasets could impact the comparison and conclusions. It would be interesting but not immediately essential to test whether cordycepin responsive genes change their APA profile in a nucleosome mutant.

According to their GEO upload, the cells used by Kaplan et al. 2009, in the nucleosome occupancy data were also BY4741. We had intended to further test the connection between chromatin architecture and cordycepin induced APA but were defeated by the pandemic. It is of clear interest to further studies.